# Long-range transported continental aerosol in the eastern North Atlantic: three multiday event regimes influence cloud condensation nuclei

**Francesca Gallo[1,a], Janek Uin[2], Kevin J. Sanchez[3], Richard H. Moore[3], Jian Wang[4], Robert Wood[5], Fan Mei[6], Connor Flynn[7], Stephen Springston[2], Eduardo B. Azevedo[8], Chongai Kuang[2], and Allison C. Aiken[1]**

[1]Earth and Environmental Sciences Division, Los Alamos National Laboratory, Los Alamos, NM, USA
[2]Environment and Climate Science Department, Brookhaven National Laboratory, Upton, NY, USA
[3]NASA Langley Research Center, Hampton, VA, USA
[4]Center for Aerosol Science and Engineering, Department of Energy, Environmental and Chemical Engineering, Washington University in St. Louis, St. Louis, MO, USA
[5]Department of Atmospheric Sciences, University of Washington, Seattle, WA, USA
[6]Atmospheric Measurement and Data Sciences, Pacific Northwest National Laboratory, Richland, WA, USA
[7]School of Meteorology, University of Oklahoma, Norman, OK, USA
[8]Group of Climate, Meteorology and Global Change (CMMG), University of Azores, Angra do Heroísmo, Portugal
[a]now at: NASA Langley Research Center, Hampton, VA, USA

**Correspondence:** Allison C. Aiken (aikenac@lanl.gov) and Francesca Gallo (francesca.gallo@nasa.gov)

**Abstract.** The eastern North Atlantic (ENA) is a region dominated by pristine marine environment and subtropical marine boundary layer clouds. Under unperturbed atmospheric conditions, the regional aerosol regime at the ENA CE1 varies seasonally due to different seasonal surface-ocean biogenic emissions, removal processes, and meteorological regimes. However, during periods when the marine boundary layer aerosol at the ENA is impacted by particles transported from continental sources, aerosol properties within the marine boundary layer change significantly, affecting the concentration of cloud condensation nuclei (CCN). Here, we investigate the impact of long-range transported continental aerosol on the regional aerosol regime at the ENA using data collected at the U.S. Department of Energy's (DOE) Atmospheric Radiation Measurement (ARM) user facility on Graciosa Island in 2017 during the Aerosol and Cloud Experiments in the Eastern North Atlantic CE2 (ACE-ENA) campaign. We develop an algorithm that integrates number concentrations of particles with optical particle dry diameter ($D_p$) between 100 and 1000 nm, single scattering albedo, and black carbon concentration to identify multiday events (with duration >24 consecutive hours) of long-range continental aerosol transport at the ENA. In 2017, we detected nine multiday events of long-range transported particles that correspond to $\sim 7.5\%$ of the year. For each event, we perform HYSPLIT 10 d backward trajectories analysis, and we evaluate CALIPSO aerosol products to assess, respectively, the origins and compositions of aerosol particles arriving at the ENA. Subsequently, we group the events into three categories, (1) mixture of dust and marine aerosols, (2) mixture of marine and polluted continental aerosols from industrialized areas, and (3) biomass burning aerosol from North America and Canada, and we evaluate their influence on aerosol population and cloud condensation nuclei in terms of potential activation fraction and concentrations at supersaturation of 0.1 % and 0.2 %. The arrival of plumes dominated by the mixture of dust and marine aerosol at the ENA in the winter caused significant increases in baseline $N_{tot}$. Simultaneously, the baseline particle size modes and CCN potential activation fraction remained almost unvaried, while cloud condensation nuclei concentrations increased proportionally to $N_{tot}$.

Events dominated by a mixture of marine and polluted continental aerosols in spring, fall, and winter led to a statistically significant increase in baseline $N_{tot}$, a shift towards larger particular sizes, a higher CCN potential activation fractions, and cloud condensation nuclei concentrations of $>170\%$ and up to $240\%$ higher than during baseline regime. Finally, the transported aerosol plumes characterized by elevated concentration of biomass burning aerosol from continental wildfires detected in the summertime did not statistically contribute to increase baseline aerosol particle concentrations at the ENA. However, particle diameters were larger than under baseline conditions, and CCN potential activation fractions were $>75\%$ higher. Consequentially, cloud concentration nuclei concentrations increased by $\sim 115\%$ during the period affected by the biomass burning events. Our results suggest that, through the year, multiday events of long-range continental aerosol transport periodically affect the ENA and represent a significant source of CCN in the marine boundary layer. Based on our analysis, in 2017, the multiday aerosol plume transport dominated by a mixture of dust and marine aerosol, a mixture of marine and polluted continental aerosols, and biomass burning aerosols caused increases in the $N_{CCN}$ baseline regime of, respectively, 6.6 %, 8 %, and 7.4 % at SS 0.1 % (and, respectively, 6.5 %, 8.2 %, and 7.3 % at SS 0.2 %) at the ENA.

## 1 Introduction

Atmospheric aerosols are one of the key components of the climate system interacting with clouds and affecting cloud radiative properties, height, and water content (Twomey, 1974; Albrecht, 1989). Remote marine low-lying cloud regions are thought to be the most affected by changes in aerosol properties because clouds are optically thin and the background aerosol concentration is low (Moore et al., 2013; Rosenfeld et al., 2014; Wood et al., 2015). However, the interactions among marine boundary layer (MBL) aerosol number concentration ($N_{tot}$), cloud condensation nuclei (CCN), and cloud droplet concentration under different aerosol loading are still poorly understood and remain one of the largest sources of uncertainties in climate models and future climate projections (Bony, 2005; Carslaw et al., 2013; Fan et al., 2016; Seinfeld et al., 2016).

Over the past years, an increased number of studies and field campaigns have been dedicated to remote marine low-cloud systems in the North Atlantic Ocean to improve the parametrization of aerosol and cloud processes in the MBL (Albrecht et al., 1995; Rémillard et al., 2012; Wood et al., 2015; Behrenfeld et al., 2019; Sorooshian et al., 2020; Redemann et al., 2021; J. Wang et al., 2021). The observations collected have provided invaluable insights into the potential role of aerosols in controlling cloud properties and precipitation. Namely, perturbations in aerosol properties have been found to be associated with strong synoptic meteorological variability (Rémillard et al., 2012), variations in CCN number concentrations ($N_{CCN}$) and cloud optical depth (Liu et al., 2016), and increases in larger longer-lasting cloud cover, precipitation suppression, and cooling (Rosenfeld et al., 2014). Further efforts have been focused on examining the influence of long-range transport of continental particles on unperturbed aerosol marine regimes. These studies underline the potential of long-range transported aerosols of continental origins to alter the concentration of aerosols, cloud condensation nuclei, and cloud droplets and the efficiency of precipitation formation (Garrett and Hobbs, 1995; Dadashazar et al., 2021; Tomlin et al., 2021; J. Wang et al., 2021). Despite the importance of this topic, a quantitative understanding of the cloud condensation nuclei budget changes over the North Atlantic Ocean as a function of aerosol perturbations due to continental emissions is still missing, and the aerosol indirect forcing remains uncertain (Carslaw et al., 2013).

With the goal of characterizing aerosol and cloud interactions in extratropical marine environments, in 2013, the U.S. Department of Energy's (DOE) Atmospheric Radiation Measurement (ARM) user facility established a long-term fixed site facility in the eastern North Atlantic (ENA) (Mather and Voyles, 2013; Dong et al., 2014; Logan et al., 2014; Feingold and McComiskey, 2016), in the Azores Archipelago. The ENA ARM site is located on the remote Graciosa Island, one of the smallest and least populated islands of the archipelago. Variations in synoptic meteorological conditions and the entrainment of transported continental aerosol particles from the free troposphere into the marine boundary layer periodically affect the local conditions in the Archipelago throughout the year. These features make the ENA ARM site well-suited for collecting open ocean representative measurements and an excellent location to investigate the impact of long-range transport of continental particles on low-cloud systems in pristine marine regions (Wood et al., 2015; J. Wang et al., 2021).

The ENA facility includes an aerosol observing system (AOS) for the continuous measurements of aerosol physical, optical, and chemical properties and the associated meteorological parameters at time resolutions from seconds to minutes (Uin et al., 2019). In situ AOS observations provide an unprecedented opportunity to robustly study the interaction between aerosols and clouds to achieve a quantitative understanding of the key controlling processes that drive aerosol properties and the CCN budget in the MBL. In addition to the AOS routine measurements, during two intensive operating

periods (IOPs) (June–July 2017 and January–February 2018) of the ARM Aerosol and Cloud Experiments in the Eastern North Atlantic CE3 (ACE-ENA) field campaign, the ARM Aerial Facility (AAF) Gulfstream-159 (G-1) research aircraft flew over the ENA site and provided in situ characterizations of the marine boundary layer and lower free troposphere structure, as well as the vertical distribution and horizontal variability of low clouds and aerosols (J. Wang et al., 2021). High correlation (slope $= 1.04 \pm 0.01$ TS1, $r^2 = 0.7$) between AOS submicron number concentrations of particles at the ENA fixed site and AAF measurements were found during the summer, indicating the broader regional representativeness of the AOS surface measurements when the boundary layer is well mixed (Gallo et al., 2020).

The recent results from the ACE-ENA campaigns have advanced the knowledge of aerosol process (Zawadowicz et al., 2021; J. Wang et al., 2021; Zheng et al., 2021) and cloud structures and processes (Gao et al., 2020; Yeom et al., 2021) in the remote MBL as well as allowed the evaluation of algorithms for remote sensing retrievals (Wu et al., 2020). However, many mechanisms underlying aerosol–cloud interactions over the North Atlantic remain unresolved. Within the ACE-ENA scientific objectives yet to be addressed, the complete understanding of the key controlling processes that shape the CCN budget in the MBL is critical (J. Wang et al., 2021). Motivated by this need, in this study, we leverage the AOS datasets collected at the ENA during the entire year 2017 to constrain the influence of long-range transported particles with different continental origins on the cloud condensation nuclei concentrations in the pristine marine environment. First, we develop an algorithm that integrates aerosol property indicators of the presence of continental particles to detect multiday ($>24$ consecutive hours) transported aerosol plume events at the ENA. Changes in specific aerosol properties caused by the arrival of continental air masses over the ENA region have been described in previous literature. Namely, increased concentrations of submicron aerosol particles have been reported in the western and eastern North Atlantic by a number of previous studies (Garrett and Hobbs, 1995; Logan et al., 2014; Pennypacker and Wood, 2017; Sanchez et al., 2022). Simultaneously, elevated levels of black carbons (BC) and low submicron single scattering albedo (SSA) values in different locations in the North Atlantic region have been associated with the presence of continental air masses containing products from incomplete fossil fuel combustion and biomass burning (Kleefeld, 2002; Junker et al., 2006; Costabile et al., 2013; O'Dowd et al., 2014; China et al., 2015; Cavalli et al., 2016). Based on these studies, we develop our algorithm and define specific thresholds for each of the aerosol parameters discussed above to detect periods affected by continental air masses (Sect. 2.2).

Once the multiday aerosol plume transport events have been detected by the algorithm, we assess aerosol regimes at the ENA under both regional aerosol baseline conditions (Sect. 3.1) and during the period of times impacted by the arrival of continental aerosol particles (Sect. 3.2). Namely, we first evaluate aerosol sources and sinks under unperturbed marine conditions providing the necessary framework to understand the influence of continental transport on marine aerosol population and CCN budget. Subsequently, we determine the origins and types of aerosols transported at the ENA during the multiday events using HYSPLIT backward trajectories and Cloud-Aerosol Lidar and Infrared Pathfinder Satellite Observations (CALIPSO) classification, respectively, and we quantitatively assess the influence of the events on aerosol properties at the ENA through statistical analysis. In Sect. 3.2 we present three case studies representative of the diverse continental aerosol plumes arriving at the ENA through the year: a mixture of marine aerosols and dust (Sect. 3.2.1), a mixture of polluted continental and marine aerosols (Sect. 3.2.2), and biomass burning aerosols (Sect. 3.2.3). In addition, we provide a summary statistic of multiday aerosol plume transport event influences on aerosol physical properties, such as variation in particle number concentrations and shifts in size distribution, and CCN potential activation factor and concentrations at the ENA (Sect. 3.2.4).

With this study, we aim to provide key observational constraints to parametrize the influence of changes in baseline $N_{tot}$ and particle size modes due to aerosol perturbation events on CCN regimes. Ultimately, our results might be used as a proxy to estimate the CCN budget over remote oceans and to inform climate models improvements and validation.

## 2 Measurements and methodology

### 2.1 ENA ARM facility

Measurements of in situ aerosol properties examined in this study were collected though the aerosol observing system (AOS) at the ENA ARM fixed facility on Graciosa Island (39°5′28″ N, 28°1′36″ W) (Bullard et al., 2017; Uin et al., 2019; Uin and Smith, 2020), between 1 January 2017 and 31 December 2017. The ENA ARM AOS comprises of one container that samples aerosols using instrumentations connected to a central not-heated inlet located approximately 10 m above ground. A list of the AOS measurements analyzed here, including references for each instrument, is given in Table 1 and summarized in the following sections. The pressure and temperature for the aerosol data is the same as reported in the ARM data archive which is given at ambient conditions to allow for direct comparison with the other data at the ENA. If desired for future global comparison, data can be calculated at standard temperature and pressure (STP) using the AOS meteorology data. All of the data are collected, ingested, and quality controlled by the U.S. DOE ARM user facility (Peppler et al., 2008; Uin et al., 2019). Calibrations were completed in accordance with the ARM instrument handbooks. No abnormalities were found during the periods that would affect the data reported here.

**Table 1.** Aerosol observing system measurements at the ENA ARM site analyzed in this study.

| Measurement | Symbol | Unit | Instrument | Reference |
|---|---|---|---|---|
| Submicron aerosol number concentration | $N_{tot}$ | cm$^{-3}$ | Condensation Particle Counter CPC Model 3772, TSI Inc. | Kuang and Mei (2019) |
| Size distribution of submicron aerosols (70 to 1000 nm) | | cm$^{-3}$ | Ultra-High-Sensitivity Aerosol Spectrometer UHSAS, DMT | Uin et al. (2016a) |
| Number concentration of cloud condensation nuclei | CCN | cm$^{-3}$ | Cloud Condensation Nuclei Counter CCN Model CCN-100, DMT | Roberts and Nenes (2005); Rose et al. (2008); Uin et al. (2016b) |
| Aerosol growth factor | | | Humidified Tandem Differential Mobility Analyzer HTDMA Model 3002, Bretchel | Lopez-Yglesias et al. (2014); Uin et al. (2016c) |
| Aerosol absorption coefficients | $B_{abs}$ | Mm$^{-1}$ | Particle Soot Absorption Photometer PSAP 3-λ, Radiant Research | Bond et al. (1999); Virkkula et al. (2005); Virkkula (2010); Springston (2018) |
| Aerosol scattering coefficients | $B_{sca}$ | Mm$^{-1}$ | Integrating Nephelometer Neph, Model 3563, TSI | Costabile et al. (2013); Uin et al., (2016d) |
| Non-refractory sulfate and organic | | μm$^{-3}$ | Aerosol Chemical Speciation Monitor Aerodyne Research | Ng et al. (2011); Watson, 2017) |

Prior to conducting any data analysis, periods impacted by local aerosol events ($\sim 23\%$ of the 2017 datasets used in the study) were removed from submicron aerosol number concentration ($N_{tot}$), size distribution, single scattering albedo, black carbon, and cloud condensation nuclei datasets using the ENA-Aerosol Mask algorithm specifically developed for the AOS measurements at ENA (Gallo et al., 2020; Gallo and Aiken, 2022).

### 2.1.1 Aerosol physical properties

Measurement of submicron particle number concentrations ($N_{tot}$) with particle diameter ($D_p$) >10 nm are made with a Condensation Particle Counter (CPC) Model 3772 (TSI, Inc., Shoreview, MN, USA) (Kuang et al., 2019). A Ultra-High Sensitivity Aerosol Spectrometer (UHSAS) (Droplet Measurement Technologies, Inc., Longmont, CO, USA) is used for sizing particles with $D_p$ between 70 and 1000 nm (Uin, 2016a). Size distributions of submicron aerosol particles are described by separating the data into three size modes: (1) Aitken (At) mode aerosols with $D_p \leq 100$ nm, (2) accumulation (Ac) mode aerosols with $D_p$ between 100 and 300, and (3) large accumulation (LA) mode aerosols with $D_p$ between 300 and 1000. The number concentration of the accumulation ($N_{Ac}$) and large accumulation ($N_{LA}$) mode aerosols are directly measured by the UHSAS, while CPC and UHSAS measurements are combined to calculate the Aitken ($N_{At}$) mode as the difference between total par-

ticle number concentrations and the sum of the two larger modes: $N_{At} = N_{tot} - (N_{Ac} + N_{LA})$. Number concentrations of cloud condensation nuclei ($N_{CCN}$) are measured using a Cloud Condensation Nuclei (CCN) Counter (Droplet Measurements Technologies Inc.) at five levels of supersaturations from 0.1% to 1% (Roberts and Nenes, 2005; Rose et al., 2008; Uin, 2016b). Here, we utilize CCN measurements collected at the determined supersaturation (SS) levels of 0.1% and 0.2% which represent typical maximum supersaturations in marine boundary layer clouds where CCN activation occurs (Korolev and Mazin, 2003; Clarke and Kapustin, 2010; Wood, 2012). Furthermore, we combine CPC and CCN measurements to calculate the aerosol potential activation fraction (AF) as the ratio of $N_{CCN}$ to the total submicron aerosol number. Finally, the hygroscopicity of aerosol particles with initial dry size from 50 to 250 nm is measured using a Humidified Tandem Differential Mobility Analyzer (HTDMA) (Brechtel Manufacturing, Inc.) (Uin, 2016c). Particle hygroscopic growth (HG) at subsaturated conditions is calculated as the ratio of the geometric mean mobility diameter of the humidified particles ($d_m(RH)$) (RH >85%) to the dry diameter ($d_d$) (RH between 6.1% and 7.3%). According to the kappa-Köhler theory (Petters and Kreidenweis, 2007) and using HG, we calculate the hygroscopicity parameter $\kappa$ for dry particles with $D_p = 50, 100, 150, 200,$ and 250 nm as

$$\kappa = \left(HG^3 - 1\right) \left[ \frac{\exp\left(\frac{A}{HG\,d_d}\right)}{RH} - 1 \right], \tag{1}$$

where $A$ is the Kelvin parameter defined as

$$A = \frac{4\,\sigma_w\,M_w}{RT\,\rho_w},\tag{2}$$

$M_w$, $\sigma_w$, and $\rho_w$ are, respectively, the molar mass, the surface tension, and the density of the water. $R$ is the universal gas constant, and $T$ is the temperature. The instrument and its mode of operation are described in detail by (Lopez-Yglesias et al., 2014).

### 2.1.2 Aerosol optical and chemical properties

Aerosol absorption coefficients ($B_{abs}$) are measured at ENA using a 3-wavelength Particle Soot Absorption Photometer (PSAP) at $\lambda$ of 464, 529, and 648 nm. The instrument is described in detail by Bond et al. (1999) and Virkkula et al. (2005). The response of the PSAP is affected by mass flow calibration, filter loading, the amount of light scattered by the particles, the flow rate, and the spot size of the sample (Bond et al., 1999; Virkkula et al., 2005; Virkkula, 2010). ARM archive PSAP data include corrections for the mass flow calibration and filter loading (Springston, 2018). Aerosol scattering coefficients ($B_{sca}$) at ENA are measured at $\lambda$ of 450, 550, and 700 nm using a TSI integrating nephelometer (TSI, model 3563) (Uin, 2016d). ARM archive nephelometer data include corrections for truncation and illumination errors (Uin, 2016d). Prior to measurement, the PSAP and nephelometer sample air passes through an impactor that periodically switches between 1 and 10 μm cut-point sizes (Uin et al., 2019). The $B_{abs}$ and $B_{sca}$ values discussed in this study refer to measurements collected at 1 μm cut-point sizes. The $B_{sca}$ at 450 nm was scaled to the measured $B_{abs}\lambda$ of 464 through interpolation based on the scattering Angstrom exponent (SAE) (Costabile et al., 2013). In this study we use aerosol light absorption ($B_{abs}$) and scattering ($B_{sca}$) coefficients to calculate the single scattering albedo (SSA) at 464 nm, defined as $SSA = (B_{sca})/(B_{abs} + B_{sca})$. Equivalent black carbon (BC) concentrations are estimated from ($B_{abs}$) with an assumed mass-absorbing cross section of 6.4 m$^2$ g$^{-1}$ at 648 nm (Bond and Bergstrom, 2006). Bulk particle composition measurements of the mass concentrations of non-refractory sulfate and organics are provided by an Aerodyne research aerosol chemical speciation monitor (ACSM) (Ng et al., 2011; Watson, 2017).

## 2.2 Multiday transported aerosol plume event identification algorithm and statistical analysis

We developed an algorithm to detect multiday transported aerosol plume events, which relies on the variations of physical and optical aerosol properties caused by long-range transport of particles in the eastern North Atlantic. The application of the algorithm requires continuous measurements of the following three parameters: number concentrations of particles with optical particle dry diameter ($D_p$) between 100 and 1000 nm, submicron SSA at 464 nm wavelength, and black carbon concentration. The measurements are averaged over 6 h periods which are sufficiently short to detect variations in mass properties but also sufficiently long to remove the effect of hourly variations due to diurnal cycles and processes that occur on a small timescale (Wood et al., 2017; Dadashazar et al., 2021) and match the time frequency of the HYSPLIT backward trajectories discussed below (the utilization of 7 and 8 h periods was also tested and lead to the same results). The thresholds for the three aerosol parameters are established based on earlier works conducted in the eastern North Atlantic region that describe their variations during the period affected by transport of continental air masses. Namely, Pennypacker and Wood (2017) observed at ENA daily median number concentrations of $D_p$ 100 to 1000 nm particles above 100 cm$^{-3}$ during periods dominated by high sea-level pressure and large-scale subsidence with air masses originating from North America approaching the Azores from the northwest. In the same study, the high median concentration of particles $D_p$ 100–1000 nm regime was found to be associated with median and 75th percentile SSA values of 0.92 and 0.95, respectively, at 470 nm wavelength. Black carbon concentrations ranging between 10 and 40 ng m$^{-3}$ during clean conditions have been reported by field studies conducted in different locations in the North Atlantic (O'Dowd et al., 2004; Shank et al., 2012; Cavalli et al., 2016). Quinn et al. (2019) and Sakerin et al. (2021) have reported average BC concentrations between 15 and 25 ng m$^{-3}$ and 37 and 44 ng m$^{-3}$, respectively, in the western North Atlantic during the NAAMES field campaigns and during cruise expeditions conducted between 2007 and 2020 over the North Atlantic Ocean. A threshold of 75 ng m$^{-3}$ has been typically utilized to indicate the presence of continental influenced air masses (Cooke et al., 1997; Kleefeld, 2002; Junker et al., 2006), while Pohl et al. (2014) have used BC concentrations ranging from 20 and 44 ng m$^{-3}$ to identify clean background in the subtropical Atlantic. In more recent works, Facchini et al. (2008) and O'Dowd et al. (2014) determined BC 50 ng m$^{-3}$ as a threshold value to identify combustion influences at Mace Head. Similarly, Saliba et al. (2020) and Lawler et al. (2020) used the same criterion to separate ambient marine from continental periods in the western North Atlantic. Based on this literature, the algorithm flags the data as affected by long-range transported aerosols when the following conditions occur at the same time for at least 24 consecutive hours (four consecutive 6 h periods): (1) median number concentration of $D_p$ 100–1000 nm particles $>100$ cm$^{-3}$ over 6 h period, (2) median submicron single scattering albedo at $\lambda$ 464 nm $<0.95$, and (3) mean black carbon concentrations $>40$ ng m$^{-3}$. It is important noting that the utilization of medians instead of means for number concentrations of $D_p$ 100–1000 nm particles and SSA to constrain periods impacted by long-range transport events in Pennypacker and Wood (2017) is due to the need of minimize the potential impacts of unidentified outlier. In our study, we performed

post-data processing methods prior to conducting any data analysis to remove short-duration high-concentration aerosol events (Gallo et al., 2020), and we obtained similar mean and median values (difference between mean and median values <12 %) for the three parameters used to develop the multiday transported aerosol plume event identification algorithm. Therefore, to allow a better comparison of our results to the previous literature, the algorithm relies on the utilization of median values for number concentrations of particles with $D_p$ between 100 and 1000 nm and submicron SSA at 464 nm wavelength and of mean values for the black carbon concentration.

Once the multiday transported aerosol plume events are detected, their origins and transport paths are evaluated by performing 10 d backward trajectories arriving at 50 and 500 m above the ENA site. The analyses are conducted using the Hybrid Single-Particle Lagrangian Integrated Trajectory (HYSPLIT) 4 model (Stein et al., 2015) with a time step of 6 h using the National Center Environmental Prediction (NCEP) Global Data Assimilation System (GDAS) meteorological data and model vertical velocity as input. In addition, Cloud-Aerosol Lidar and Infrared Pathfinder Satellite Observations (CALIPSO) aerosol products within the first 1500 m of the vertical column (corresponding to the mean MBL depth over mid-latitude ocean (Rémillard et al., 2012)) are used, when available, to assess the predominant types of aerosol particles arriving at ENA during the events (Omar et al., 2009). CALIPSO classification includes six types of aerosol mixtures: clean continental, clean marine, dust, polluted continental, polluted dust, and smoke (Burton et al., 2013). Finally, to assess the correlation between origin and composition of the multiday transport events and their influence on baseline aerosol properties at ENA, we perform a post hoc Tukey–Kramer honest significant difference (HSD) test (Haynes, 2013) to determine whether the arrival of the continental aerosol plumes produced statistically significant changes on (a) baseline aerosol number concentrations ($\Delta N_{tot}$), (b) aerosol mode sizes in terms of relative Aitken and accumulation mode contributions to $N_{tot}$ (expressed as the ratio between $N_{At}$ and $N_{Ac}$ ($\Delta N_{At}/N_{Ac}$)), and (c) CCN potential activation fraction ($\Delta AF$). The significance probability was assessed at the probability level of $p < 0.05$, and statistical analyses were performed using Igor Pro 8 with the Statistic package (WaveMetrics Inc.).

# 3 Results and discussion

The entrainment of continental particles from long-range transport represents a significant source of aerosols over mid-latitude oceans and has the potential of altering the regional aerosol regimes (Garrett and Hobbs, 1995; Honrath, 2004; Roberts et al., 2006; García et al., 2017; Zhang et al., 2017; Zheng et al., 2018). Here, we apply the algorithm to detect multiday transported aerosol events at the ENA during the

year 2017. Measurements affected by local aerosol events were removed prior the application of the algorithm following Gallo et al. (2020). Once the events had been identified, we removed the measurements affected by the arrival of continental aerosol plumes and we extracted the aerosol baseline conditions (period of time not affected by local aerosol events and/or long-range transported plumes) to assess the aerosol seasonal regimes at ENA (Sect. 3.1). Subsequently, the multiday aerosol plume transport events are examined and categorized based on origin and composition and their impacts on aerosol physical properties, such as variation in particle number concentrations and shifts in size distribution, which affects the ability of particles to act as CCN are evaluated (Sect. 3.2).

## 3.1 Regional aerosol regime under baseline conditions

### 3.1.1 Concentration and size distribution of submicron aerosol particles

The concentration of submicron aerosol particles and their size distribution under baseline conditions at the ENA shows seasonal variations likely related to a combination of different regional emission sources and sink mechanisms. In remote marine regions like the ENA, particles of marine origin, including sea spray aerosols and marine aerosols formed by biogenic volatile organic compounds produced by marine phytoplankton, dominate the aerosol population in the marine boundary layer (Rinaldi et al., 2010; Lapina et al., 2011; Sanchez et al., 2018). Overall, we found lower concentrations of submicron particles in the winter (January–February, and November–December 2017) and higher during late spring and summer (Fig. 1). Namely, the minimum monthly $N_{tot}$ mean value was observed in January 2017 ($260 \pm 143 \, \text{cm}^{-3}$), while the maximum monthly $N_{tot}$ mean value was reached in June 2017 ($523 \pm 259 \, \text{cm}^{-3}$), approximately 2 times the winter minimum. Our results are consistent with earlier studies and field campaigns conducted in the North Atlantic Ocean region which report low wintertime $N_{tot}$ as the result of reduced contribution from ocean biological activities and higher occurrence of in-cloud precipitation and coalescence scavenging during winter months compared to the spring and summer (Pennypacker and Wood, 2017; Zheng et al., 2018; Quinn et al., 2019; Gallo et al., 2023). Likewise, the concentration of particles in the Aitken and accumulation modes follows similar seasonal trends with monthly mean minima in $N_{At}$ and $N_{Ac}$ in January 2017 ($N_{At} = 148 \pm 81 \, \text{cm}^{-3}$) and in November ($N_{Ac} = 90 \pm 53 \, \text{cm}^{-3}$), respectively, and maxima in June 2017 ($N_{At} = 360 \pm 97 \, \text{cm}^{-3}$ $N_{Ac} = 195 \pm 79 \, \text{cm}^{-3}$) (Fig. 1). Interestingly, we observed that summer (June–September 2017) mean $N_{Ac}$ values, which are approximately double those in the winter, are considerably higher than the correspondent median $N_{Ac}$ values (Fig. 1). There is minimal influence of local aerosol sources on accumulation mode

aerosol measurements at the ENA and the data utilized here have been filtered to remove the impact of potential local emissions (Gallo et al., 2020). However, in the summer, MBL baseline aerosol concentrations might be influenced by the entrainment of diluted and aged continental particles from the free troposphere, which likely contributes to enhanced concentration of particles in the accumulation mode (J. Wang et al., 2021). This observation is consistent with previous studies investigating aerosol vertical profiles during the summer ACE-ENA field campaign (J. Wang et al., 2021) and over the western North Atlantic during the NASA North Atlantic Aerosol and Marine Ecosystems Study campaign (NAAMES). Particles in the large accumulation mode (not shown) showed the opposite seasonal trend, reaching the maximum monthly mean value in the winter ($N_{LA} = 14 \pm 9 \, \text{cm}^{-3}$ in January) and the lowest concentrations in the summer ($N_{LA} = 7 \pm 4 \, \text{cm}^{-3}$ in August). However, throughout the entire year, the total aerosol number concentration among the three particle modes is dominated by the Aitken mode (yearly mean Aitken mode contribution to $N_{tot} = 61 \% \pm 3 \%$), while the accumulation mode is lower (yearly mean Accumulation mode contribution to $N_{tot} = 35 \% \pm 4 \%$), and the large accumulation mode represents only a small percentage of $N_{tot}$ (yearly mean large accumulation mode contribution to $N_{tot} = 3 \% \pm 1 \%$). Further analysis of the measured size distribution from the UHSAS instrument (measurement size range 70–1000 nm) during winter (January, February, November, and December 2017) and summer (May to September 2017) at the ENA provide an insight into seasonal variations of particle size. In the wintertime, mean particle size $D_p$ peaks at 128 nm (Fig. 2a), while in the summer mean mode $D_p$ shifts towards slightly larger sizes, peaking at 147 nm (Fig. 2b). While the UHSAS lower size limit is at $D_p = 70$ nm, the UHSAS size distribution measurements associated with the calculated $N_{At}$, $N_{Ac}$, and $N_{La}$ suggest aerosol bimodal structure for both winter and summer. In the absence of the entrainment of particles of continental origins, the size distribution of particles in the MBL is shaped by different seasonal surface-ocean biogenic emissions, aerosol removal processes, and meteorological regimes (Behrenfeld et al., 2019). New particle formation events in the upper part of the decoupled MBL have been reported by previous studies and are due to a combination of reduced existing aerosol surface area, passage of cold fronts, reactive gas availability, and high actinic fluxes (Bates et al., 1998; Kolstad et al., 2009; Zheng et al., 2021). In the winter, more frequent precipitation and drizzle at ENA lead to the removal of large particles, such as sea spray aerosols, and consequently low existing aerosol surface availability, which are associated with wintertime cold temperature and enhance the occurrence of new particle formation events. Once formed, the new particles grow into larger particles, strongly contributing to $N_{At}$. The removal of Aitken mode particles is largely driven by coagulation, while the condensational growth into Ac mode is weak due to

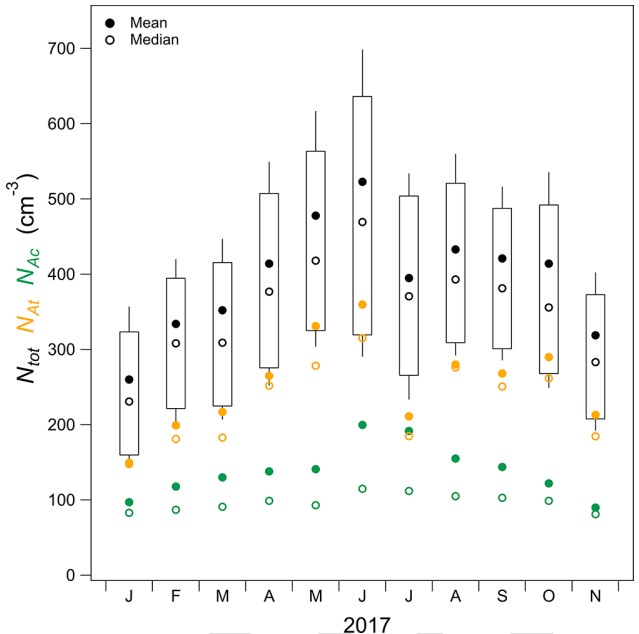

**Figure 1.** Box and whisker plot of monthly submicron aerosol number concentrations (box bottom at 25 %, box top at 75 %, whisker bottom at 10 %, and whisker top at 90 %). Mean (circles) and median (open circles) of total number concentration (black), number of Aitken (yellow), and accumulation (green) modes.

low dimethyl sulfide (DMS) concentrations in the MBL and only represents a minor source of MBL $N_{Ac}$ (Zheng et al., 2018). On the contrary, sea spray aerosol production at the surface ocean due to enhanced wintertime wind speeds up to 21.7 m s$^{-1}$ (Aiken et al., 2019) substantially contributes to large accumulation mode concentrations, explaining the higher $N_{La}$ observed in January (Vignati et al., 2010; Zheng et al., 2018; Quinn et al., 2019). During late spring and summer, the phytoplankton bloom is responsible for strong ocean emissions of dimethylsulfide, whose oxidation products have been found to enhance the condensational growth of nucleation mode particles into the Aitken and subsequently to the Ac modes (O'Dowd et al., 1997; Andreae et al., 2003; Zheng et al., 2018). Furthermore, photochemistry and/or oxidation of oxygenated gas-phase organic compounds of marine origin produce secondary organic aerosols at the surface layer which contribute to the growth of Aitken mode particles during late summer when phytoplankton activity is lower (Mungall et al., 2017). In a previous study conducted at the ENA between 2015 and 2018, Zheng et al. (2018) assessed the correlations between wind speeds and particle size. In the summer, no correlations between wind speeds and $N_{At}$ and $N_{Ac}$ were reported, while $N_{LA}$ was observed to strongly correlate with wind speeds, therefore suggesting that the contribution from sea spray is limited to the large accumulation mode.

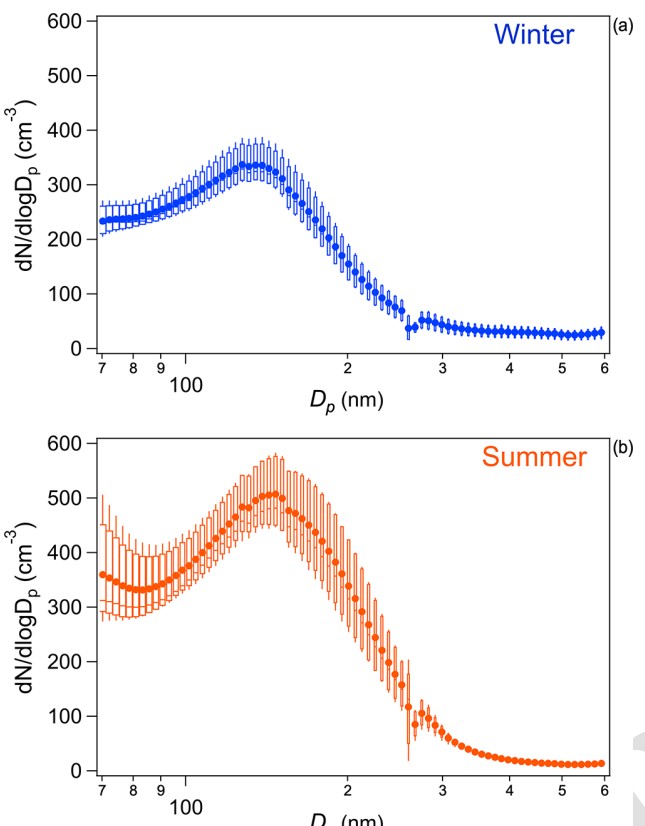

**Figure 2.** Particle size distribution in winter **(a)** and summer **(b)** at ENA. The whiskers and boxes show the 90th, 75th, median, 25th, and 10th percentiles, and the circles represent the mean value per each size bean circle. Discontinuity is at around 270 nm due to technical limitations of the UHSAS (handoff region between two internal gain stages). CE4

### 3.1.2 CCN concentrations and potential activation fraction

The concentration of CCN in the remote marine boundary layer is dominated by ocean-derived particles. Previous studies have reported that the major sources of CCN over the Atlantic Ocean include sea salt aerosols enriched in organics and marine biogenic gases that oxidize and condense onto existing particles (Charlson et al., 1987; Pandis et al., 1994; O'Dowd et al., 2004; Yoon et al., 2007; Korhonen et al., 2008; Quinn and Bates, 2011; Sanchez et al., 2018; Zheng et al., 2018). Here, we assess the seasonal variations of CCN concentrations ($N_{CCN}$) at the ENA under baseline conditions, and we investigate the CCN potential activation fractions to evaluate how the different aerosol seasonal regimes affect the ability of the particles to act as CCN.

Throughout the year 2017, mean monthly CCN concentration values were low, as expected for clean marine environments (Ovadnevaite et al., 2014), and seasonal variations were noticeable at both super-saturations (Fig. 3). Lower monthly mean $N_{CCN}$ values were reported in the winter and spring (minimum in December 2017 and $N_{CCN,0.1\%} = 69 \pm 27\,cm^{-3}$ at SS of 0.1 and $N_{CCN,0.2\%} = 108 \pm 38\,cm^{-3}\,cm^{-3}$ at SS 0.2 %), while monthly $N_{CCN}$ mean values were higher in the summer (maximum monthly $N_{CCN}$ mean values in July 2017 and were $141 \pm 53\,cm^{-3}$ at SS of 0.1 and $178 \pm 68\,cm^{-3}$ at SS 0.2 %). The CCN potential activation fraction follows a different seasonal trend, exhibiting higher values in late summer/fall and winter (mean AF SS 0.1 % was $0.27 \pm 0.03$ and mean AF SS 0.1 % was $0.41 \pm 0.02$) and lower values in the spring (mean AF SS 0.1 % was $0.22 \pm 0.01$ and mean AF SS 0.2 % was $0.32 \pm 0.04$). The low number particle concentration and consequently low concentrations of cloud condensation nuclei observed in the MBL can be to a large degree attributable to reduced ocean biological activity in the winter. Furthermore, CCN removal through in-cloud coalescence scavenging processes associated with high occurrence of precipitation events in the winter and spring might also play a role in constraining CCN concentrations (Sharon et al., 2006; Zheng et al., 2018; Sanchez et al., 2022). However, the higher CCN potential activation fraction in wintertime than in the spring indicates that winter aerosol particles have a more elevated ability to act as cloud condensation nuclei. Supporting our finding, J. Wang et al. (2021) reported a high precipitation rate and an increase in CCN coalescence scavenging, accompanied by enhanced $N_{Ac}$ activation at the ENA during the ACE-ENA winter field campaign. A slightly lower ratio of $N_{Ac}$ to $N_{tot}$ in the winter than in the summer (mean accumulation mode ratio to $N_{tot} = 31\%$ and 37 %, respectively, in the winter and in the summer) suggests that particle compositions play an important role in CCN formation at the ENA. Consistent with our observations, earlier studies have pointed out that wind-generated sea spray aerosols enriched by particulate organic matter and biogenic sulfate, as observed at ENA in the winter are a stronger source of CCN than aerosols generated by phytoplankton activities at the surface ocean. (Quinn and Bates, 2011; Sanchez et al., 2018; O'Dowd et al., 2004). The higher summertime $N_{CCN}$ observed here are in agreement with previous studies conducted at the ENA, which also found a correlation between elevated $N_{CCN}$ and concentration of cloud droplet (Wood et al., 2015; J. Wang et al., 2021) and reduced precipitation (Rémillard and Tselioudis, 2015; Giangrande et al., 2019), thus suggesting minimal CCN removal through wet scavenging. Simultaneously, strong volatile organic compound (VOC) emissions at the surface ocean due to the final phase of the phytoplankton bloom and microbial activities lead to the formation of highly hygroscopic secondary sulfate particles which grow quickly into CCN by condensation and well explain the elevated $N_{CCN}$ and potential activation fractions found here (Saliba et al., 2020; Zawadowicz et al., 2021).

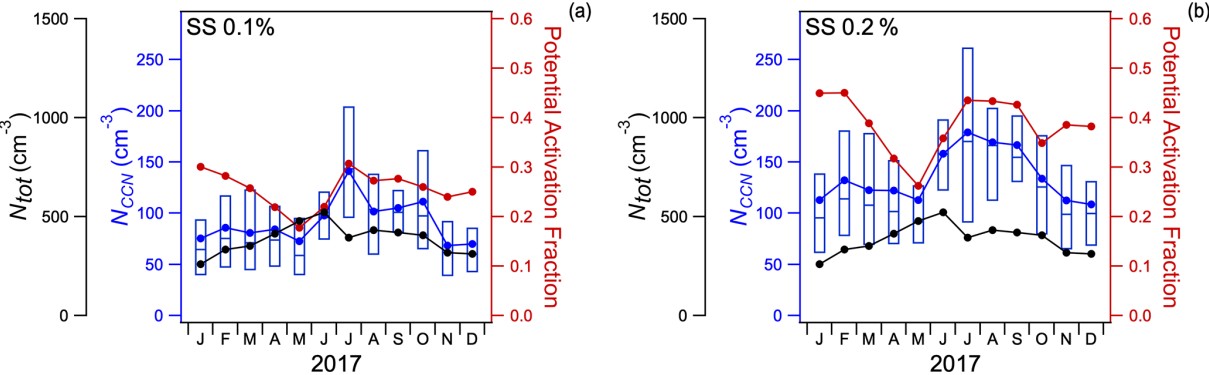

**Figure 3.** Box and whisker plot of $N_{CCN,0.1\%}$ (**a**) and $N_{CCN,0.2\%}$ (**b**), mean $N_{CCN}$ is the blue circles, median -, box bottom at 25 %, box top at 75 %, whisker bottom at 10 %, and whisker top at 90 %, mean $N_{tot}$ is the black circles, and CCN potential activation fraction is the red circles.

## 3.2 Multiday transported aerosol plume events at ENA in 2017

We apply the algorithm for detecting multiday aerosol plume transport events at the ENA to the in situ aerosol measurements collected at ENA ARM during 2017. We identify nine events affecting the ENA through the entire year. The duration of the events was typically greater than 2 d, with an average time period of 3 d, and a total duration of 642 h corresponding to $\sim 7.5\%$ of the year. A summary of the events, including duration, origins, aerosol CALIPSO classification, and values of the three aerosol properties used to identify the events (median number concentration of $D_p$ 100–1000 nm particles, mean single scattering albedo of submicron particles at $\lambda$ 464 nm, and mean black carbon concentration) is shown in Table 2.

The origin of the air masses arriving at the ENA and their paths, assessed by performing 10 d HYSPLIT backward trajectories, indicates a seasonal pattern likely controlled by seasonal meteorological regimes and atmospheric circulation in the Northern Hemisphere (Zhao et al., 2012). A number of studies reported Sahara dust intrusions into the North Atlantic MBL in the late fall and winter associated with cyclonic dust storms in the North Africa region (Nakamae and Shiotani, 2013; Choobari et al., 2014; Laken et al., 2014; Logan et al., 2014; Cuevas et al., 2017). Similarly, air masses from the Arctic might also represent a source of mineral dust at the ENA in the spring (Zheng et al., 2018). In the case of the Arctic, the atmospheric load of dust particles is the result of bare soil surface and glacial outwash plains, and it is projected to increase over the next years as a consequence of the retreat of glaciers (Bullard et al., 2016; Tobo et al., 2019). In accordance with these observations, we found two events of southward transport from northern African and Portuguese flows to the ENA in the months of November and December 2017, likely favored by Arctic anticyclone, polar vortex, and midlatitude circulation and an event of transport from Arctic in March 2017. A large fraction of air masses arriving at the ENA throughout the year are attributed to transport from industrialized continental areas such as North Europe, Canada, and North America due to midlatitudes cyclones and convection (García et al., 2017). Continental aerosol particles are emitted in the boundary layer by anthropogenic processes and are subsequently transported for several days within the free troposphere before entrainment into the marine boundary layer over the North Atlantic Ocean (Honrath, 2004; Wood et al., 2015; Cavalli et al., 2016). Here, we observed two aerosol transport events with northern European origins in the months of January and April 2017. We also identified aerosol transport events from North America and Canada between May and September 2017. Our results are consistent with previous studies conducted in the North Atlantic region, which reported dominant eastward direction from North America in the late spring and summers (Zhao et al., 2012) and high aerosol loading due to pollution outflow and biomass burning emissions (Honrath et al., 2004; Alves et al., 2007; Dzepina et al., 2015; García et al., 2017; Zheng et al., 2020; J. Wang et al., 2021)

Analyses of CALIPSO aerosol products provide further insights on the type of aerosols transported. Consistent with the origin of the emission sources, we observed dust and marine aerosol mixtures associated with transport from the Arctic and Canada in March 2017 and from North Africa in November and December 2017, while the airflows originating from industrialized areas (January, April, May, and October 2017) typically consisted of a mixture of polluted continental aerosol, smoke, and marine particles. Simultaneously, NASA Worldview VIIRS 375 observations of the multiday aerosol plume transport events occurring in August and September 2017 show an elevated concentration of smoke over the ENA due to a particularly intense wildfire season in North America and Canada, therefore suggesting the presence of biomass burning aerosols as also previously observed by Zheng et al. (2020).

**Table 2.** Summary of multiday transported aerosol plumes events that affected the ENA in 2017 including duration, aerosol emission origins, and CALIPSO classification. The values of the three aerosol properties used by the algorithm to detect the events (median concentration of particles with $D_p$ 100–1000 nm, median SSA 1 μm at λ 464 nm, and mean BC values) are shown in the rightmost column during each event (first line) and under baseline condition (in italic, second line).

| Event | Duration (hours) | Origin (HYSPLIT) | CALIPSO aerosol classification | Median concentration particles $D_p$ 100–1000 nm | Median SSA 1 μm (λ 464 nm) | Mean BC (ng m$^{-3}$) |
|---|---|---|---|---|---|---|
| 7 to 11 January | 114 | Northern Europe | Mixture of dust, polluted continental aerosols, and smoke | 365 cm$^{-3}$ *83 cm$^{-3}$* | 0.87 *0.96* | 229 ± 41 ng m$^{-3}$ *36 ± 21 ng m$^{-3}$* |
| 12 to 15 March | 72 | Arctic/Canada | Mixture of dust and marine aerosols | 319 cm$^{-3}$ *91 cm$^{-3}$* | 0.93 *0.96* | 115 ± 37 ng m$^{-3}$ *35 ± 19 ng m$^{-3}$* |
| 20 to 22 April | 54 | Northern Europe | Mixture of marine and polluted continental aerosols and smoke | 460 cm$^{-3}$ *99 cm$^{-3}$* | 0.94 *0.95* | 121 ± 27 ng m$^{-3}$ *29 ± 21 ng m$^{-3}$* |
| 21 to 22 May | 36 | North America | Polluted continental aerosol and smoke | 608 cm$^{-3}$ *93 cm$^{-3}$* | 0.94 *0.97* | 142 ± 16 ng m$^{-3}$ *33 ± 20 ng m$^{-3}$* |
| 26 to 29 August | 84 | North America | Elevated smoke | 332 cm$^{-3}$ *105 cm$^{-3}$* | 0.94 *0.95* | 181 ± 58 ng m$^{-3}$ *40 ± 25 ng m$^{-3}$* |
| 9 to 13 September | 96 | North America/Canada | Data not available | 289 cm$^{-3}$ *103 cm$^{-3}$* | 0.93 *0.96* | 175 ± 39 ng m$^{-3}$ *39 ± 22 ng m$^{-3}$* |
| 11 to 13 October | 48 | Hurricane Ophelia | Mixture of dust, marine and polluted continental aerosols, and smoke | 329 cm$^{-3}$ *99 cm$^{-3}$* | 0.89 *0.96* | 144 ± 69 ng m$^{-3}$ *30 ± 19 ng m$^{-3}$* |
| 26 to 28 November | 54 | North Africa | Mixture of dust and marine aerosols | 271 cm$^{-3}$ *81 cm$^{-3}$* | 0.91 *0.96* | 181 ± 29 ng m$^{-3}$ *34 ± 21 ng m$^{-3}$* |
| 7 to 10 December | 84 | North Africa | Mixture of dust and marine aerosols | 235 cm$^{-3}$ *80 cm$^{-3}$* | 0.92 *0.96* | 103 ± 18 ng m$^{-3}$ *26 ± 18 ng m$^{-3}$* |

Finally, through statistical analysis we were able to correlate the aerosol plumes origin, their composition, and the influences that they exert on $N_{tot}$ and particle size seasonal regime at the ENA to group the multiday transport events with similar characteristics into the following three categories: (1) dust and marine mixture events – including the March 2017 event with Arctic and Canadian origins and the November and December 2017 events from North Africa, which caused a statistically significant increase in baseline $N_{tot}$ and statistically non-significant shifts in baseline size distribution and CCN potential activation fractions; (2) polluted continental and marine mixture – including January, April, May, and October 2017 events originating in continental industrialized areas, which caused statistically significant changes in baseline submicron particle number concentration, baseline size distribution, and baseline CCN potential activation fractions; (3) biomass burning – including the August and September 2017 events, which caused statistically non-significant changes in baseline submicron aerosol particles but did produce statistically significant shifts in baseline particle size distribution and an increase in the CCN potential activation fraction.

In the following three sections, we discuss case studies representative of the diverse continental plumes arriving at the ENA through the year, while in Sect. 3.2.4, Table 3, and Fig. 8 we provide a summary statistic of the three multiday event regimes mentioned above.

### 3.2.1 Multiday transport event of dust and marine mixture aerosols from North Africa

The transport of air masses from North Africa to the North Atlantic Ocean region during the winter is the result of the shift of the subtropical high pressure system southeastward and enhancing trade winds over the Sahara (Chiapello, 2005; Riemer et al., 2006; Alonso-Pérez et al., 2011; Nakamae and Shiotani, 2013). Sahara dust intrusions in the North Atlantic MBL have been reported by a number of studies (Choobari et al., 2014; Laken et al., 2014; Cuevas et al., 2017), especially between January and March (Alonso-Pérez et al., 2007). During the transport over the ocean, dust particles typically mix with marine aerosols (Peshev et al., 2019), undergoing heterogenous chemical reactions and removal mechanisms that alter their composition and size and, as a consequence, their influence on the CCN aerosol baseline regime. In this study, we identified the arrival of air masses from the western Sahara and Mauritania to the ENA between 7 and 12 December 2017 (Fig. 4). Here, we assess CALIPSO retrievals, aerosol hygroscopicity parameters as a function of dry particle size ($\kappa_{HTDMA}$), non-refractory sulfate and organic mass, concentrations of black carbon, and CO to confirm the nature of the aerosol particles arriving at the ENA during the event (Fig. 7c). CALIPSO aerosol profiles indicate the presence of a mixture of dust and marine aerosol in the marine boundary layer. Simultaneously, $\kappa_{HTDMA}$ values were 0.22, 0.30, 0.37, 0.32, and 0.37, respectively, for dry particles with $D_p = 50$,

**Table 3.** Summary of the characteristics of each type of multiday aerosol plume transport event. Italic values indicate statistically significant $\Delta$.

| | Dust and marine mixture | Polluted continental and marine mixture | Biomass burning |
|---|---|---|---|
| Events date (year 2017) and origin | 12 to 15 March – Arctic/Canada<br>26 to 28 November – North Africa<br>7 to 10 December – North Africa | 7 to 11 January – North Europe<br>20 to 22 April – North Europe<br>21 to 22 May – North America<br>11 to 13 October – hurricane | 26 to 29 August – North America<br>9 to 13 September – North America |
| Statistical analysis | | | |
| $\Delta N_{tot}$ | *>110 %* | *Between 95 % and 110 %* | *<25 %* |
| $\Delta N_{At}/N_{Ac}$ | *<1 %* | *>200 %* | *>200 %* |
| $\Delta AF_{0.1\%}$ | $\sim 5\%$ SS 0.1 % | *Between 30 % and 75 %* | *>75 %* |
| $\Delta AF_{0.1\%}$ | $\sim 7\%$ SS 0.2 % | *Between 35 % and 100 %* | *>75 %* |
| Size mode fraction | | | |
| $N_{At}$ contribution to $N_{tot}$ | $\sim 59\%$ | $\sim 42\%$ | $\sim 33\%$ |
| $N_{Ac}$ contribution to $N_{tot}$ | $\sim 38\%$ | $\sim 56\%$ | $\sim 63\%$ |

100, 150, 200, and 250 nm (Fig. 7a). For representative atmospheric aerosol particles, the hygroscopicity parameter $\kappa_{HTDMA}$ ranges from 0 to 1.4, where high values ($>0.5$) indicate very hygroscopic inorganic species such as sodium chloride, and low values indicate non-hygroscopic organic enriched compounds ($0.01 < \kappa_{HTDMA} < 0.5$ slightly to very hygroscopic, and $\kappa_{HTDMA} < 0.01$ non-hygroscopic components) (Petters and Kreidenweis, 2007). Although freshly emitted Sahara dust particles are typically not soluble, depending on the transport path and environmental conditions during the transport, heterogenous chemical interactions with other atmospheric particles and trace gases can influence their composition and enhance their hygroscopicity (Levin, 2005; Kallos et al., 2007; Astitha et al., 2010). The $\kappa_{HTDMA}$ values observed here were accompanied by mean sulfate and organic mass concentrations, respectively, 1.63 and 0.91 µg m$^{-3}$, corresponding to 7-fold and 2-fold increase, respectively, in sulfate and organic masses compared to the baseline regime during the month of December 2017, suggesting that sulfates of marine and anthropogenic origins likely coat the dust, making the particles more hygroscopic (Fig. 7c) (Koehler et al., 2009; Choobari et al., 2014; Zhang et al., 2014). Mean black carbon concentrations during the event were also higher than for the rest of the month (event mean BC was $101 \pm 17$ and up to 120 ng m$^{-3}$ against baseline mean BC in December 2017 which was $26 \pm 8$ ng m$^{-3}$), while CO levels remained constant (event mean CO was $101.4 \pm 3$ ppmv against baseline mean CO in December 2017 which was $100.9 \pm 9$ ppmv). Consistent with our results, previous studies found that aerosol from biomass burning activities occurring during the dry season in the Sahel region (Boreal winter) can mix with the dust, affecting the composition of the particles without the transport of smoke over the Atlantic (Ben-Ami et al., 2009; Redemann et al., 2021). The arrival of the aerosol plume at the ENA was associated with an increase in mean submicron aerosol number concentration of approximately double that under baseline [CE5] conditions (mean event $N_{tot} = 683 \pm 135$ cm$^{-3}$ compared to monthly mean $N_{tot}$ December 2017 was $313 \pm 128$ cm$^{-3}$). Aitken and accumulation mode particle concentrations both double, while the relative contributions of the two modes to $N_{tot}$ remained similar to baseline with mean Aitken contribution of 59 % and mean accumulation contribution of 38 % of $N_{tot}$ ($N_{At}/N_{Ac}$ change $= 0.3\%$) (Fig. 7b), indicating that the particles arriving at the ENA during the event had a size distribution similar to that of the regional aerosol. The peak of the size distribution in the accumulation mode was at 127 nm for both event and baseline aerosol regimes, while the concentration of $D_p > 200$ nm particles was only 11 % of $N_{Ac}$. Our results are in good agreement with previous studies conducted over the central Atlantic Ocean (Astitha et al., 2010) and in the Cape Verde region (Formenti et al., 2003) which found a high number concentration of particles in the Aitken mode, associated with the arrival of a mixture of dust and anthropogenic sulfate from North Africa. Namely, Formenti et al. (2003) reported $N_{At}/N_{Ac}$ ratio $\sim 1.5$–3 and size distribution dominated by particles with $D_p$ 150 nm. The size of dust particles over the Atlantic Ocean is the result of a combination of different source regions, dust generation mechanisms, atmospheric synoptic conditions, and sink mechanisms. A number of previous studies have focused on the evolution of the size distribution of dusty aerosols during transport over the North Atlantic and report rapid loss of coarse mode particles due to gravitational settling and wet deposition just off the coast of Africa, while finer particles remain in suspension and can be transported for longer distances (Maring, 2003; Kalashnikova and Kahn, 2008; Lawrence and Neff, 2009; Mahowald et al., 2014; Friese et al., 2016) [CE6]. The concentration of CCN increased during the event following a similar trend of $N_{tot}$ (mean event $N_{CCN} = 70 \pm 27$ and $109 \pm 31$ cm$^{-3}$ compared to monthly mean $N_{CCN}$ December 2017 which was $165 \pm 32$ and $280 \pm 36$ cm$^{-3}$, re-

spectively, for SS 0.1 % and 0.2 %, and corresponding increases by factors of 2.3 and 2.5, respectively, over baseline value observed during the month of December 2017) leading to almost no change in CCN potential activation fraction (event AF was 0.25 and 0.42 compared to AF in December 2017 which was 0.26 and 0.42, respectively, for SS 0.1 % and 0.2 %). Furthermore, the linear regression between $N_{tot}$ and $N_{CCN}$ during the event and under baseline conditions shows similar slopes (at SS 0.1 %: $N_{CCN} = 0.23N_{tot}$, and $N_{CCN} = 0.18N_{tot}$, respectively, during the event and under baseline conditions at SS 0.2 %: $N_{CCN} = 0.40N_{tot}$ and $N_{CCN} = 0.30N_{tot}$ during the event and under baseline conditions, respectively), indicating that the enhanced concentration of CCN observed during the event is mainly due to higher $N_{tot}$ (Fig. 7d). Furthermore, when comparing the potential activation fraction to the ratio of $N_{Ac}$ and $N_{At}$ we observed a good linear regression (AF $= 0.42N_{Ac}/N_{At}$, with $r^2 = 0.93$ at SS 0.1 %, and AF $= 0.67N_{Ac}/N_{At}$, with $r^2 = 0.90$ at SS 0.2 %), suggesting a strong correlation between CCN activation and particle size (Fig. 7e).

### 3.2.2 Multiday transport event of polluted continental and marine mixture aerosols from North Europe

Air masses from the Arctic and Europe occasionally reach the northeast Atlantic during the spring months (Zheng et al., 2018), while transport from this region during summer and winter is rare (Zhang et al., 2017). Here, we describe a transport event of marine and polluted continental aerosol mixture at the ENA which occurred between 20 and 22 April 2017. The aerosol plume originated from the Arctic and, before entraining into the MBL at ENA, traveled for several days over northern Europe (Fig. 5). CALIPSO aerosol retrievals indicate the presence of a mixture of marine and polluted continental aerosols (Table 2). Typically, during the transport to the ENA, air masses are contaminated by industrial and urban pollution over industrialized European regions. The source apportionment of aerosol in Europe has been examined in previous studies by means of long-term studies and long-term station and satellite retrievals (Ng et al., 2010; Yang et al., 2020; Bressi et al., 2021; Chen et al., 2022). Over Europe, the major contribution to aerosol emissions in central and northern Europe is by particles from solid fuel combustion with both residential and urban/industrial origins (Karagulian et al., 2015; Thunis et al., 2018), while biomass burning from wild fires (Pio et al., 2008) and agricultural fires only contributes marginally in eastern Europe (Stohl et al., 2002 TS2). As a result, non-refractory sulfates, primary organic aerosols, and BC are emitted in the atmosphere, leading to average annual concentrations (period including years 2014 to 2018) over Europe of 1.80, 0.94, and 0.23 µg m$^{-3}$, respectively (Yang et al., 2020). However, the types of emission sources and aerosol contributions vary seasonally, leading to higher aerosol mass concentrations in the wintertime and lower in the summertime (Yang

et al., 2020; Chen et al., 2022). Typically, freshly emitted urban/industrial particles are the result of incomplete combustion processes and consist of soot and hydrophobic organic compounds that do not show high hygroscopic growth (Swietlicki et al., 2008). However, once in the atmosphere, photochemical aging processes and changes in mixing state (e.g., coating of hydrophilic material) increase particle hygroscopicity (Weingartner et al., 1997 TS3; Wang et al., 2010) and their ability to act as CCN (Wittbom et al., 2014). Here, we observed $\kappa_{HTDMA}$ values almost constantly across the measured particle size range of 50 to 250 nm ($\kappa_{HTDMA}$ values were 0.44, 0.44, 0.49, 0.48, and 0.49, respectively, for dry particles with $D_p = 50$, 100, 150, 200, and 250 nm), which suggests the presence of aged, well-mixed particles (Fig. 7f). Mass concentrations of non-refractory sulfate and organics were, respectively, 1.03 and 0.50 µg m$^{-3}$ and were almost 3-fold and 5-fold higher than during the baseline regime (mean sulfate and organic concentrations in April 2017 were 0.36 and 0.11 µg m$^{-3}$, respectively) (Fig. 7h). Furthermore, mean BC concentration were $121 \pm 33$ and up to 176 ng m$^{-3}$ during the time period affected by the transport of particles from northern Europe and higher than what was observed during baseline conditions (monthly mean in April 2017 was $36 \pm 16$ cm$^{-3}$ ng m$^{-3}$ TS4), which also confirmed the presence of particles with urban and industrial origins. CO levels were also slightly higher than under baseline conditions, ranging between 120 and 135 ppb (baseline CO concentration $<112$ ppb) indicative of a moderately polluted boundary layer (Spackman et al., 2011 TS5). A statistically significant increase in $N_{tot}$ baseline regime was observed during the event (mean $N_{tot}$ event was $804 \pm 155$ cm$^{-3}$ against monthly mean $N_{tot}$ April 2017 which was $414 \pm 124$ cm$^{-3}$). The accumulation mode particle concentration was 3-fold higher during the event than under baseline conditions with the size distribution peaking between 135 and 140 nm, while the increase in $N_{At}$ was statistically significantly lower ($= +25$ %) (Fig. 7g). Consequently, the mean particle diameter shifted toward larger sizes and the contribution of the accumulation mode to $N_{tot}$ became predominant over the Aitken mode (At contribution was 40 %, accumulation contribution was 57 %, corresponding to change in $N_{At}/N_{Ac} = 148$ %). During the event, $N_{CCN}$ exhibited mean values of $179 \pm 45$ cm$^{-3}$ at SS 0.1 % (compared to the monthly mean of April 2017 which was $84 \pm 37$ cm$^{-3}$) and $379 \pm 23$ cm$^{-3}$ at SS 0.2 % (compared to the monthly mean of April 2017 which was $122 \pm 67$ cm$^{-3}$). The total CCN active fraction was also statistically significantly higher during the event than under the baseline regime, being 30 % at SS 0.1 % and 49 % at SS 0.2 % and corresponding to a 34 % and 53 % increase at SS 0.1 % and SS 0.2 %, respectively. Accordingly, the slopes of the linear regression between $N_{tot}$ and $N_{CCN}$ are higher during the event than under baseline conditions (at SS 0.1 %: $N_{CCN} = 0.28N_{tot}$ and $N_{CCN} = 0.19N_{tot}$, respectively, during the event and under baseline conditions, and at SS 0.2 %: $N_{CCN} = 0.46N_{tot}$ and $N_{CCN} = 0.28N_{tot}$ during

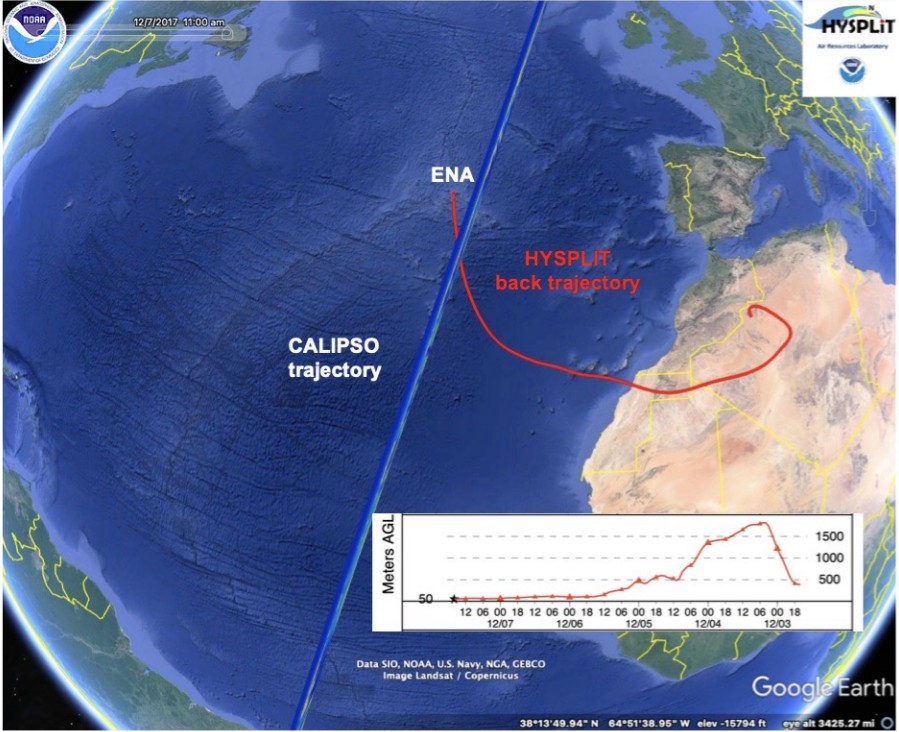

**Figure 4.** CALIPSO trajectories (blue) and HYSPLIT back trajectories (red) arriving at 50 m a.g.l. above the ENA site on 7 December 2017 (© Google Earth 2015).

the event and under baseline conditions, respectively) indicating the enhanced ability of the continental transported particles to act as CCN (Fig. 7i, j TS6). While comparing the potential activation fraction to the ratio of $N_{Ac}$ and $N_{At}$, the shape of the curves generated were different at SS 0.1 % and 0.2 % (Fig.7j). A linear regression was generated at SS 0.1 % (AF $= 0.17 + 0.07 N_{Ac}/N_{At}$, with $r^2 = 0.83$) versus a lognormal distribution at SS 0.2 %. These results suggest that, while at lower supersaturation the number of activated particles was mainly driven by a shift towards larger particle size; at higher supersaturation particle composition also played a strong role. Thus, the increase in $N_{CCN}$ during the event was likely trigged by the combination of high $N_{tot}$, elevated relative contribution of accumulation mode particles to $N_{tot}$, and high $\kappa_{HTDMA}$ values. Supporting our findings, previous studies have hypothesized that shortly after emitted in the atmosphere, sulfate particles mix with BC and other inorganic and organic species. As a consequence, during the transport particles can reach larger $D_p$ and become more hygroscopic due to the presence of sulfate in the mixture, therefore enhancing the CCN active fraction (Swietlicki et al., 2008; Massling et al., 2015).

### 3.2.3 Multiday transport event of biomass burning aerosols from North America

Pollution and biomass burning aerosols from North America commonly impact the remote North Atlantic region (Honrath, 2004; Alves et al., 2007; Dzepina et al., 2015; García et al., 2017). Zhang et al. (2017) reported that 16 %, 15 %, and 13 % of the air masses intercepted at Pico Mountain, respectively, in spring, summer, and fall are influenced by North American anthropogenic sources with 7.3 % being associated with wildfire influences. Namely, during summer 2017, several severe wildfires raged in the United States and northwest Canada (Kloss et al., 2019). Biomass burning particles in the smoke from the wildfires are typically released into the lower extratropical stratosphere and transported by cold fronts through the jet stream eastward over the Atlantic Ocean, where cold descending airstreams favor their entrainment in the MBL (Owen et al., 2006; Khaykin et al., 2018; Peterson et al., 2018). Here we present a detailed characterization of a long-range transport event of biomass burning aerosols that affected the ENA between 9 and 13 September 2017. During this period, the arrival of air masses from North America are associated with an elevated number of active wildfires in North America and Canada, as observed by NASA Worldview VIIRS 375 active fires counts between 1 and 15 September 2017. (Fig. 6). Long-range transported biomass burning aerosols from North Ameri-

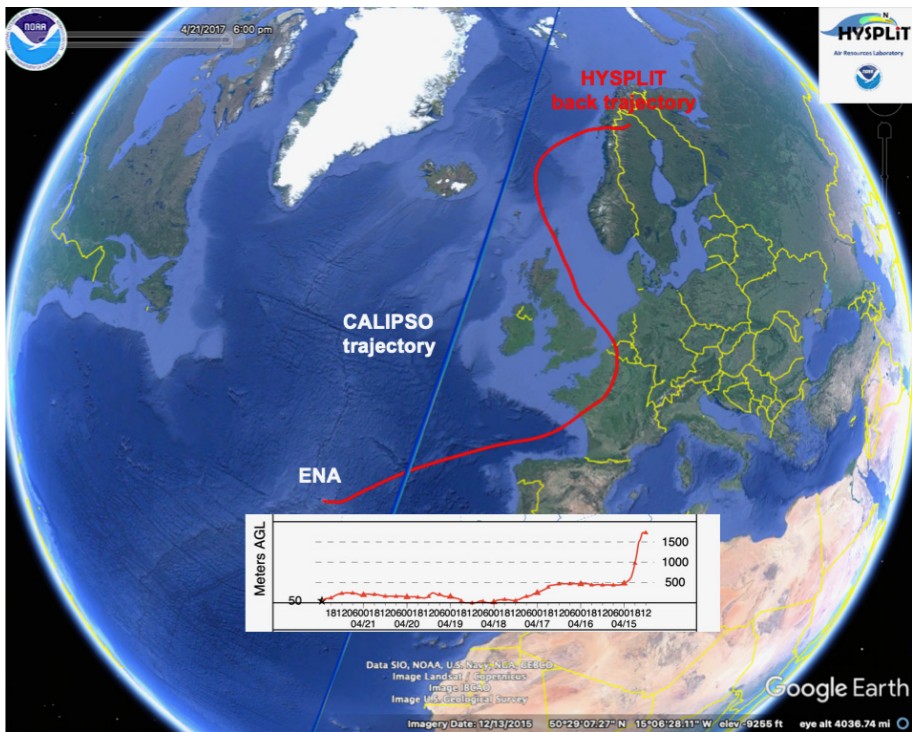

**Figure 5.** CALIPSO trajectories (blue) and HYSPLIT back trajectories (red) arriving at 50 m a.g.l. above the ENA site on 21 April 2017 (© Google Earth 2015).

can and Canadian wildfires at the ENA in August 2017 have also been reported by a previous study (Zheng et al., 2020). The presence of biomass burning particles is confirmed by the analysis of aerosol optical properties (data not shown), which shows mean aerosol absorption coefficients at $\lambda$ 648 nm $= 1.04 \pm 0.28$ Mm$^{-1}$ and mean SSA at $\lambda$ 464 $= 0.93 \pm 0.02$, in agreement with values reported by previous studies of North American aged wildfire aerosols (Clarke et al., 2007; Zheng et al., 2020), The $\kappa_{\mathrm{HTDMA}}$ values were lower than baseline conditions at 0.32, 0.31, 0.28, 0.28, and 0.29 for particles with $D_{\mathrm{p}} = 50$, 100, 150, 200, and 250, respectively, (Fig. 7k), while mean sulfate and organic concentrations were, respectively, 4-fold and 9-fold higher than under the baseline regime (being that mean sulfate and organic concentration during the event was 1.75 and 4.25 µg m$^{-3}$, respectively, and during the month of September 2017 was 0.4 and 0.46 µg m$^{-3}$, respectively) (Fig. 7m). Simultaneously, mean BC concentrations were $175 \pm 9$ ng m$^{-3}$ during the time period affected by the transport of particles from northern Europe (vs. monthly mean BC in August 2017 which was $39 \pm 22$ ng m$^{-3}$) and mean CO $= 186 \pm 64$ ppb (against mean baseline CO concentration in August 2017 which was $94 \pm 7$ ppb) was indicative of moderately polluted boundary layer. The substantially elevated concentration of organics and BC particles during the event (up to 8.65 µg m$^{-3}$ and 841 ng m$^{-3}$) explains the low hygroscopicity of the aerosol particles in the plume, as re-

ported by earlier laboratory studies on aged biomass fuel representative of North American wildfires (Petters et al., 2009; Lathem et al., 2013) (Fig. 7i). The event did not cause a statistically significant increase in particle number concentrations (mean event $N_{\mathrm{tot}} = 530 \pm 189$ cm$^{-3}$ compared to monthly mean $N_{\mathrm{tot}}$ December 2017 which was $421 \pm 139$ cm$^{-3}$), which were dominated by particles with $D_{\mathrm{p}} > 100$ nm (mean Aitken and accumulation contributions to $N_{\mathrm{tot}} = 37$ % and 58 %, respectively) (Fig. 7l). Fresh biomass burning aerosol commonly has a unimodal distribution with $D_{\mathrm{p}}$ between 30 and 100 nm (Hosseini et al., 2010; Levin et al., 2010). However, during the transport events, aerosol processes such as coagulation and condensation of organic material onto existing particles lead to the formation of larger particles with $D_{\mathrm{p}}$ between 170 and 300 nm and to narrower size distribution compared to that of freshly emitted particles (Zellner, 2000; Dentener et al., 2006; Janhäll et al., 2010). Associated with the above-mentioned shift in size distribution, we found potential activation fractions (0.44 and 0.70 at SS 0.1 % and 70 % at SS at 0.2 %, respectively) approximately twice those under baseline conditions, suggesting that the transported aerosol particles are more effective as CCN. Similarly, the $r^2$ and slopes obtained through linear regression between $N_{\mathrm{tot}}$ and $N_{\mathrm{CCN}}$ are higher under periods affected by the events compared to baseline conditions ($r^2 = 0.56$ with a slope of $0.44 \pm 0.005$ at SS 0.1 % and $r^2 = 0.66$ with a slope of $0.68 \pm 0.007$ at SS 0.2 % during the event, against $r^2 = 0.32$

with a slope of $0.22 \pm 0.001$ at SS 0.1 % and $r^2 = 0.40$ with a slope of $0.34 \pm 0.007$ at SS 0.2 %) (Fig. 7n). The CCN concentration was higher (respectively, 220 % at SS 0.1 % and 227 % at SS 0.2 %) during the event then for the rest of the month of September 2017. Furthermore, comparing the potential activation fraction to the ratio of $N_{Ac}$ and $N_{At}$, we obtained lognormal distributions at both SS 0.1 % and SS 0.2 %, indicating that particle composition also affects the concentration of particles that can act as CCN (Fig. 7o). These results demonstrate that aged wildfire aerosols dominate the accumulation mode and act better as CCN and affect CCN budget at the ENA with potential effects on Earth's albedo, cloud's lifetime, and precipitation (Albrecht, 1989).

### 3.2.4 Continental aerosol influences on regional aerosols properties and CCN

Multiday aerosol plume transport events at the ENA influence regional aerosol properties and CCN concentrations. However, the extent of changes in $N_{tot}$ and particle size mode are dependent on the origin and composition of the transported particles and affect CCN concentrations differently. Here, we provide a summary statistic of the influence of continental aerosol emissions on baseline aerosol population and baseline CCN concentrations at the ENA for the three multiday event regimes discussed in Sect. 3.2.

The arrival of the mixture of marine aerosol and dust plumes as observed in the months of March 2017, with Arctic and Canadian origins, and in November and December 2017 originated in North Africa and provoked a statistically significant increase in $N_{tot}$ (123 %), accompanied by statistically non-significant shifts in size distribution and a CCN potential activation fraction (Fig. 8a). Namely, particle concentrations in the Aitken and accumulation modes show comparable increase (mean increase was 117 % and 146 %, respectively, for $N_{At}$ and $N_{Ac}$) (Fig. 8a), and consequently the relative Aitken and accumulation mode contributions to $N_{tot}$ remain almost constant ($N_{At}/N_{Ac}$ changes <1 %), with the mean Aitken and accumulation modes being, respectively, 59 % and 38 % of the total number concentration (Fig. 8b) and similar to the baseline condition (where the At mode contributes 61 % and the accumulation mode 36 % to $N_{tot}$) (Fig. 8b). The gravitational settling of coarse particles during the transport to the ENA is likely the reason why we did not find statistically significant shifts towards larger particle sizes (Lawrence and Neff, 2009; Mahowald et al., 2014; Friese et al., 2016). Although wet scavenging might also have played a role in the removal of coarse particles. The arrival of the aerosol plumes at the ENA also leads to higher CCN concentrations (mean increase was 122 % and 162 %, respectively, at SS 0.1 % and SS 0.2 %) than under an unperturbed aerosol regime. However, these increases were not accompanied by statistically significant changes in CCN potential activation fractions, which remained similar to the baseline conditions during the entire duration of the event (mean AF during the event: $AF_{0.1\%} = 0.26$ and $AF_{0.2\%} = 0.42$ against mean AF under unperturbed aerosol conditions: $AF_{0.1\%} = 0.27$ and $AF_{0.2\%} = 0.41$) (Fig. 8a). These results indicate that a mixture of dust and marine aerosol particles has the same ability of acting as CCN that marine regional aerosol at the ENA has, and the elevated $N_{CCN}$ are a consequence of increased $N_{tot}$.

The multiday aerosol plume transport events that occurred in the months of January, April, May, and October of 2017, dominated by a mixture of marine and polluted continental aerosol and originated in continental industrialized areas such as northern Europe and North America, caused statistically significant changes in baseline submicron particle number concentration, size distribution, and CCN potential activation fraction. Furthermore, we found a $\sim$ 4-fold higher aerosol absorption coefficient at 648 nm and a mean absorption Angstrom exponent at $\lambda 460/648$ nm $= 1.04 \pm 0.1$ Mm$^{-1}$ during the events, and the mean black carbon concentration was $177 \pm 76$ and up to 319 ng m$^{-3}$ against the mean concentration under unperturbed aerosol conditions of $35 \pm 16$ ng m$^{-3}$ (data not shown), as expected for aerosols with enhanced contribution from fossil fuel and urban pollution sources (Clarke et al., 2007; Cazorla et al., 2013). During the events, the number concentration of submicron particles at the ENA experienced a mean increase of 108 % due to 37 % and 256 % mean increases, respectively, in the $N_{At}$ and $N_{Ac}$ modes (and corresponding to $N_{At}/N_{Ac}$ changes >200 %) (Fig. 8a). Therefore, the accumulation mode became predominant over the Aitken mode. Namely, during multiday aerosol plume transport events, the average contributions of the Aitken mode to the total number particle concentrations was 42 % (between 35 % and 45 % depending on the event), while the average contributions of the accumulation mode to $N_{tot}$ was 56 % (between 45 % and 60 %) (Fig. 8b). The aforementioned changes in the baseline aerosol regime in terms of number particle concentrations and shifts in size distributions caused higher CCN concentrations (mean increase was 176 % and 240 %, respectively, at SS 0.1 % and SS 0.2 %) and statistically significant increases in CCN potential activation fractions (mean AF during the event: $AF_{0.1\%} = 0.34$ and $AF_{0.2\%} = 0.55$ corresponding to AF increases between 25 % and 50 %) (Fig. 8a). This result suggests that polluted particles of continental origins with $D_p > 100$ nm are sufficiently large to readily serve as CCN and have the potential to substantially increase CCN concentrations in marine remote regions (Hudson and Xie, 1999).

Finally, the long-range transport of smoke and biomass burning aerosols identified in the months of August and September 2017 did not impact the concentration of submicron aerosol particles, causing only a weak increase ($< +$ 25 %) in submicron number particle concentrations. However, the events led to statistically significant shifts in particle size distribution and an increase in the CCN potential activation fraction with respect to baseline conditions, namely

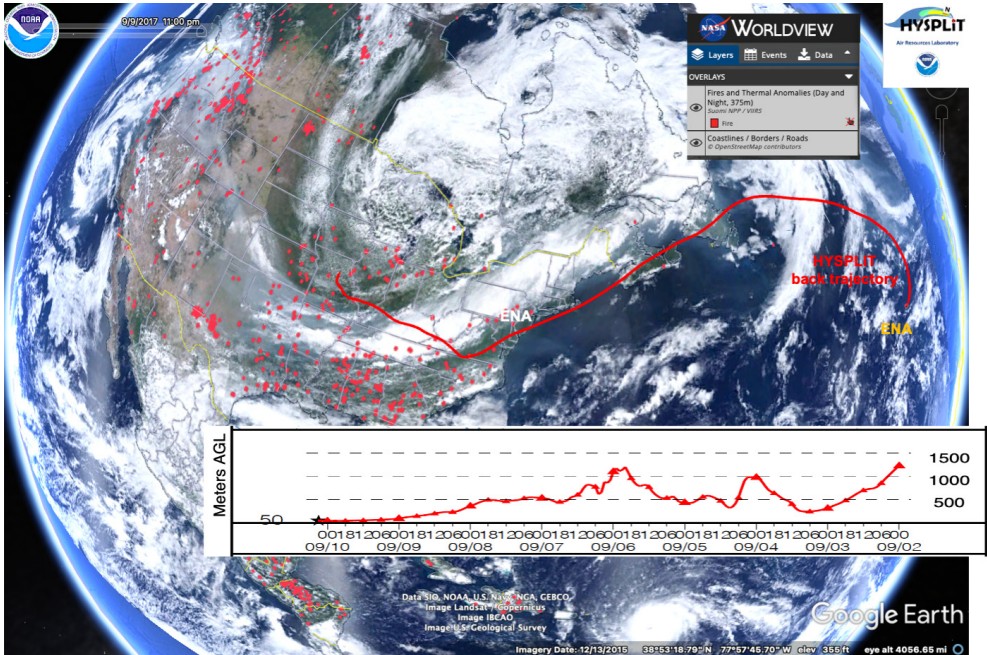

**Figure 6.** NASA Worldview VIIRS 375, active fires between 1 and 15 September 2017 (red circles) and HYSPLIT back trajectories arriving at 50 m a.g.l. above the ENA site on 10 September 2017 (© Google Earth 2015).

these events caused only a weak increase ($< +25\%$) in submicron number particle concentrations accompanied by the decrease of Aitken mode particle concentrations (mean reduction was $-39\%$ and down to $-50\%$) and the increase of accumulation mode particle concentrations (mean increase $=$ $+115\%$) (Fig. 6i, j). Thus, during the high CCN activation events, Aitken mode particles only represented 33 % of $N_{tot}$, while mean accumulation mode contribution to $N_{tot}$ was 63 % (Fig. 8b). The shift in size distribution corresponded to a decrease in the $N_{At}/N_{Ac}$ ratio of $\sim 300\%$. Simultaneously, mean CCN concentrations and AF values were 118 % and 119 % higher during the event compared to baseline conditions at SS of 0.1 % and 0.2 %, respectively, and were associated with an elevated mean CCN potential activation fraction ($= 0.46$ at SS 0.1 %, and 0.74 at SS 0.2) (Fig. 8a). These findings suggest that the shape of the submicron particle size distribution exerts a considerable effect on the ability of aerosols to act as CCN, and the arrival of biomass burning aerosols from continental wildfires statistically significantly affects the CCN concentrations at the ENA.

## 4 Conclusions

Multiday aerosol events due to long-range transport of continental aerosols are observed at the ENA throughout the year. In this study, we develop an algorithm that integrates submicron aerosol size distribution, single scattering albedo, and black carbon concentration measurements to identify multiday aerosol plume transport events occurring at the ENA in 2017. In the year 2017, we identified nine events of long-range transported particles (with durations $>24$ h), corresponding to $\sim 7.5\%$ of the year. Analysis of 10 d HYSPLIT backward trajectories and CALIPSO aerosol products indicate different origins and aerosol compositions of the air masses arriving at the ENA during the transport events. Namely, we observed the arrival of (1) a mixture of dust and marine aerosols from the Arctic and Canada in March 2017 and from North Africa in November and December 2017, (2) a mixture of marine and polluted continental aerosols from northern Europe and North America in January, April, May, and October 2017, and (3) pollution and biomass burning aerosol from North America and Canada in the months of August and September 2017. Subsequently, we assess the influence of aerosol plumes composition on CCN concentrations at the ENA, investigating the mechanisms that trigger the increase in $N_{CCN}$. The events characterized by the arrival of a mixture of dust and marine aerosols at the ENA caused statistically significant increases in $N_{tot}$, while the aerosol size distribution and CCN potential activation fraction remained similar to baseline conditions, indicating that greater $N_{CCN}$ were attributable to the elevated concentration of particles during the event. High CCN concentrations are attributed to both high $N_{tot}$ and also the dominance of particles large enough to act as CCN ($D_p > 100$ nm) from mixed marine and polluted continental aerosol plumes. Conversely, despite only causing slight increases in baseline $N_{tot}$, the events dominated by the arrival of biomass burning aerosols were characterized by the presence of particles with

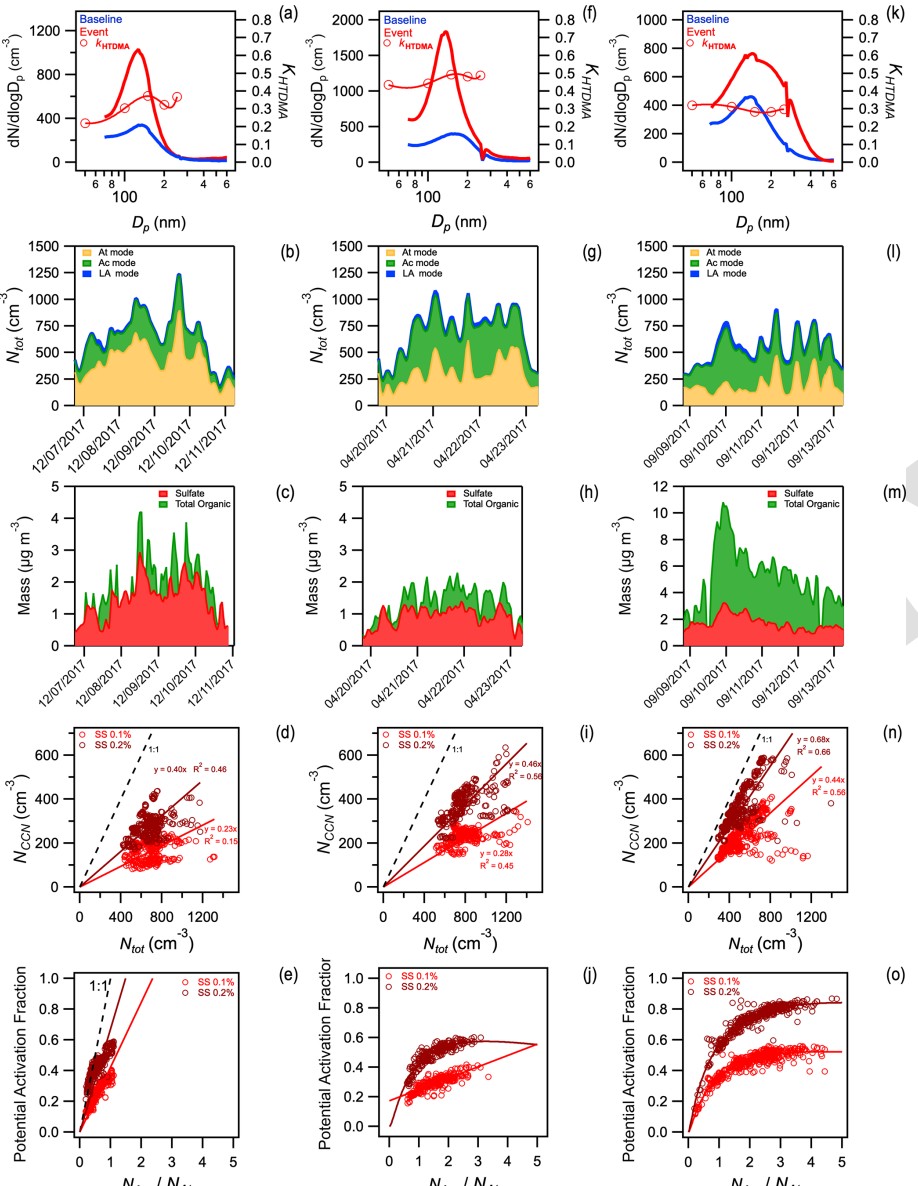

**Figure 7.** Case study of December 2017 (leftmost), April 2017 (center), and September 2017 (rightmost) events. Submicron particle size distribution under baseline conditions (blue) and during the events (red) and $\kappa_{HTDMA}$ (open circles) during the events **(a, f, k)**; Aitken, accumulation, and large accumulation mode contributions to **(b, g, l)**; non-refractory sulfate and organic aerosols **(c, h, m)**; scatter plot of $N_{CCN}$ versus $N_{tot}$ during the event (red circle) and fitting lines for the events at SS 0.1 % (red) and at SS 0.2 % (dark red) **(d, i, n)**; and plot of potential activation ratio versus $N_{Ac}/N_{At}$ or the events at SS 0.1 % (red) and at SS 0.2 % (dark red) **(e, j, o)**.

a strong ability to act as CCN, leading to 2-fold higher $N_{CCN}$. Based on our analysis, the transport of continental particles at the ENA caused a total $N_{CCN}$ increase by ∼ 22 % with respect to the CCN baseline regime, impacting ∼ 28 d and strongly contributing to the CCN concentrations at the ENA in 2017. Namely, we observed that plumes dominated by a mixture of dust and marine aerosols, a mixture of marine and polluted continental aerosols, and biomass burning aerosols can cause, respectively, a 6.5 %, 8 %, and 7.4 % increase in $N_{CCN}$. Furthermore, we showed that, once the multiday

aerosol event is identified, the analysis of changes in baseline $N_{tot}$ and particle size distribution, as well as their correlation, might be used as a proxy to estimate how CCN are affected. Based on the characteristics of the type of events discussed above, in the future an algorithm to predict $N_{CCN}$ variations during multiday events of long-range transport of aerosols could be developed and validated at the ENA to inform studies at other locations and constrain model predictions of CCN regime perturbations over remote oceans. Furthermore, the influences of aerosol perturbations on cloud properties and

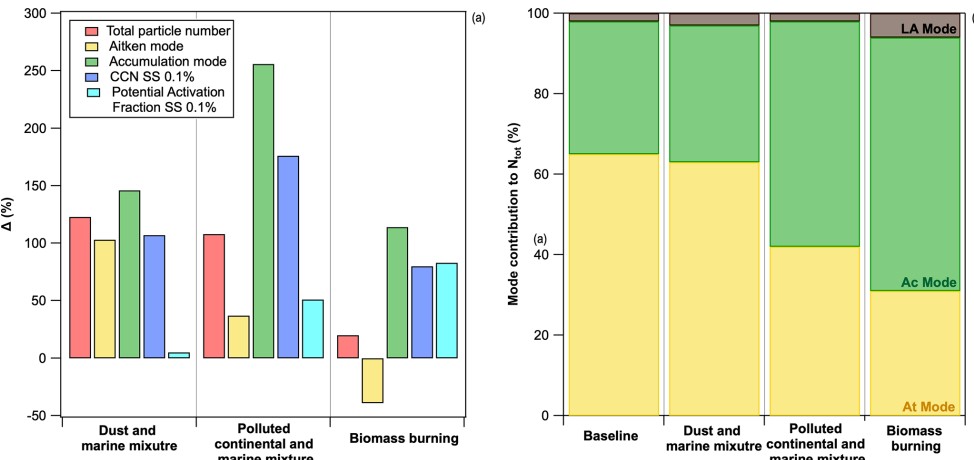

**Figure 8.** Mean percentage change in $N_{\text{tot}}$, $N_{\text{At}}$, $N_{\text{Ac}}$, $N_{\text{CCN-0.1\,\%}}$ TS7, and CCN potential activation fraction at SS 0.1 % for each type of event **(a)**; Aitken, accumulation and large accumulation particle modes relative contribution to $N_{\text{tot}}$, for baseline and each type of event **(b)**. CE7

cloud adjustments at the ENA might be explored in future studies using ARM-retrieved cloud optical properties and value-added products as well as ARM ceilometer lidar and KAZR2 datasets upon evaluations of radar and lidar techniques and the validation of retrieved observations against in situ measurements.

**Data availability.** Data used in this study are publicly accessible at the permanent archive of data collected at the eastern North Atlantic site of the Atmospheric Radiation Measurement (ARM) user facility (available at https://adc.arm.gov/discovery/#/results/ meas_category_code::aeros, last access: 8 December 2022; DOIs: https://doi.org/10.5439/1349238, https://doi.org/10.5439/1369240, https://doi.org/10.5439/1255094, https://doi.org/10.5439/1409033, https://doi.org/10.5439/1762267, https://doi.org/10.5439/1250819, https://doi.org/10.5439/1883167, Gallo and Aiken, 2022 TS8 TS9).

**Author contributions.** FG and ACA conceptualized the analysis. FG led the analyses and wrote the paper with additional input from JU and ACA. ACA was the project administrator. All authors were involved in helpful discussions and contributed to the paper.

**Competing interests.** The contact author has declared that none of the authors has any competing interests.

**Disclaimer.** Publisher's note: Copernicus Publications remains neutral with regard to jurisdictional claims in published maps and institutional affiliations.

**Special issue statement.** This article is part of the special issue "Marine aerosols, trace gases, and clouds over the North Atlantic (ACP/AMT inter-journal SI)". It is not associated with a conference.

**Acknowledgements.** The work was supported by the Atmospheric Radiation Measurement (ARM) program and funded by the U.S. Department of Energy (DOE), Office of Science, Office of Biological and Environmental Research. We acknowledge the ARM research facility, a user facility of the U.S. DOE, Office of Science, sponsored by the Office of Biological and Environmental Research, for providing data. We acknowledge the use of data and/or imagery from NASA's Fire Information for Resource Management System (FIRMS) (https://earthdata.nasa.gov/firms, last access: 8 December 2022), part of NASA's Earth Observing System Data and Information System (EOSDIS). We also acknowledge the ENA ARM site operators, Carlos Sousa, Tercio Silva, and Bruno Cunha.

**Financial support.** The work was supported by the Atmospheric Radiation Measurement (ARM) program and funded by the U.S. Department of Energy (DOE), Office of Science, Office of Biological and Environmental Research. Robert Wood was supported by the ASR award DE-SC0021103. TS10

**Review statement.** This paper was edited by Stefania Gilardoni and reviewed by two anonymous referees.

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

## Remarks from the language copy-editor

CE1  We can add "site" to some "ENA" instances as you proposed. However, it isn't clear if only the instances currently written as "at ENA" should be changed to "in the ENA" or if any other instances should also be changed. Could you please indicate which "ENA" instances should be changed to "in the ENA"? We would change those first and then add "site" to any remaining instances of "the ENA" as requested. Thank you.

CE2  Please confirm change to the official campaign name.

CE3  Please confirm change to the official campaign name.

CE4  Please confirm the changes to this caption.

CE5  Please confirm the changes to this sentence.

CE6  Please confirm the changes to this sentence.

CE7  Please confirm the changes to this caption.

## Remarks from the typesetter

TS1  Please confirm.

TS2  Please confirm citation.

TS3  Please confirm citation.

TS4  Please check unit. Is it correct as is? Or should "$cm^{-3}$" be deleted? If so, please give an explanation of why this needs to be changed. We have to ask the handling editor for approval. Thanks.

TS5  Please confirm citation.

TS6  Please confirm.

TS7  Please note: in the previous proofreading, I was hinting at the other subscripts in the text that have a comma rather than a hyphen (e.g. $N_{\mathrm{CCN},0.1\%}$). Is the subscript still correct as is?

TS8  Please confirm citation.

TS9  Please provide reference list entry for each DOI that has no citation.

TS10  Please confirm acknowledgements and financial support sections.

TS11  Please confirm reference list entry.

TS12  Please confirm title.

TS13  Please confirm article number.

TS14  Please confirm reference list entry.