# Peer review of "Long-range transported continental aerosol in the Eastern North Atlantic: three multiday event regimes influence cloud condensation nuclei"

_Atmospheric Chemistry and Physics, 2022_

## Author Comment (AC1)

**Manuscript No.**: acp-2022-637

**Title**: Long-range transported continental aerosol in the Eastern North Atlantic: three multiday event regimes influence cloud condensation nuclei

**Responses to Anonymous Referee #1**

*General Comments:*

**1. [Referee #1]**: *This paper presents novel research with a developed algorithm that utilizes several aerosol properties using thresholds from previous studies to identify and classify multi day aerosol plume transports in the Eastern North Atlantic (ENA). The authors perform statistical analysis to determine if differences in aerosol properties during regional aerosol baseline conditions and plume transport events are statistically significant. They go a step further by using HYSPLIT to determine their origin and CALIPSO to determine their type. Finally, the authors present 3 case studies corresponding to each of the 3 classification schemes. Overall, this is a good quality study with clear motivation, methodology, and discussion of results that is strongly supported by past literature. It could provide a useful constraint for climate models. It is certainly of interest for publication, although without a stronger comparison to literature to make the significance of the findings more clear, it may fit better as a Measurement Report.*

**[Resp.]:** We thank Anonymous Referee #1 for supporting our work and their well-considered comments which helped to significantly improve the manuscript. We have revised the manuscript according to Referee #1 suggestions and we have expanded the comparison of our results to the literature in the Results and discussion section to make the significance of the results clearer. We believe that the manuscript is now stronger, and more capable of higher impact providing advances in the understanding of the impact of aerosol perturbations over the North Atlantic ocean, critical for model improvements and validation. Therefore, we feel that the revised version of the manuscript has the required characteristics to be a Research Article. All the alterations to the manuscript, including improved comparison to previous studies conducted at ENA, are shown in the track changes revised version of the manuscript and all the comments are addressed in the in the following point-by-point responses below. Please, note that throughout this response, the original Referee #1 comments are highlighted in italic black and our responses follow in blue.

**2.[Referee #1]**: *The main weakness with the paper is that it seems to be attempting source attribution without chemical measurements, relying solely on back trajectories. Certainly this has been done before, but the authors would need to carefully review the success of those attempts in order to provide appropriate context for this work. However, I do wonder why this is done here, given that AOS includes ACSM measurements. Is there some problem with those that prevents their inclusion? If ACSM only available for part of the time, could that be used to strengthen the conclusions of this work by showing similarities for part of time?*

**[Resp.]:** We thank Anonymous Referee #1 for this suggestion. We have conducted further analysis using ARM ACSM data and we have incorporated our results in **Sections 3.2.1, 3.2.2, 3.2.3** and **Figures 7c, 7h, and 7m** of the revised manuscript. The discussion has been reviewed and further comparison with previous literature has been added.

[revised manuscript text omitted]

**3.[Referee #1]**: *The work discusses the algorithm for classification and how it is applied, but never actually provides the algorithm. There is a discussion of "multiday transport" criteria, but I am more interested in the differences of the 3 categories identified in abstract. Or is this just a subjective classification of 9 events based on Table 2? Table 3 provides the average characteristics of each, but if the separation is based on backtrajectories then what are the*

*specific criteria for those or are they clustered or something? Sorry if I missed it, but I assume it is not based on CALIPSO as Table 2 might indicate. Also, the CALIPSO mixtures show more complexity than the three categories in Table 3 and the abstract. Or does that result refer to just 3 case studies rather than 3 categories of the 9 events (abstract: "group the events into 3 categories")?*

[Resp.]: We thank Anonymous Referee #1 for highlighting this point and helping us streamline the text. We do agree with Referee #1 that the explanation of the criteria used to group the events is not clear through the manuscript and that without further explanation the reader might get the impression that the events were categorize merely using back trajectories and CALIPSO products. Instead, we also performed statistical analysis to determine whether the arrival of the continental aerosol plumes produced statistically significant changes in baseline 1) aerosol number concentrations, 2) aerosol mode sizes, and 3) CCN potential activation fraction. Subsequently, we used back trajectories, CALIPSO products, and the statistical analysis results as criteria to assess the correlation between plume origin, composition, and influence extent on aerosol regime at ENA to group the events with similar characteristics within three different categories. We have referred to the utilization of statical analysis in the Introduction and restructured Sections 3.2 and 3.2.4 to ensure that our criteria for grouping the events are properly explained in the revised version of the manuscript as follow:

> **Page 3, Line 39,** "[….] and we quantitatively assess the influence of the events on aerosol properties at ENA through statistical analysis. In section 3.2 of  and we present three […]"

> **Page 6, Line 34,** "Finally, to assess the correlation between origin and composition of the multiday transport events and their influence on baseline aerosol properties at ENA, we perform post hoc Tukey-Kramer Honest Significant Different (HSD) test (Haynes, 2013) determining whether the arrival of the continental aerosol plumes produced statistically significant changes on baseline a) aerosol number concentrations ($\Delta N_{tot}$), b) aerosol mode sizes in terms of relative Aitken and Accumulation modes contributions to $N_{tot}$ (expressed as the ratio between $N_{At}$ and $N_{Ac}$ ($\Delta N_{At}$ / $\Delta N_{Ac}$), and c) CCN potential activation fraction ($\Delta AF$). The significance probability was assessed at the probability level of $p < 0.05$ and statistical analyses were performed using Igor Pro 8 with Statistic package (WaveMetrics Inc.)."

> **Page 10, Line 24,** "Finally, through the statistical analysis we were able to correlate aerosol plume origin, composition, and the influences that they exert on $N_{tot}$ and particle size seasonal regime at ENA to group the multiday transport events with similar characteristics into the following three categories: 1) Dust and marine mixture events - including March 2017 event with Arctic and Canada origins, and November and December 2017 events from North Africa, which caused statistically significant increase in baseline $N_{tot}$ and statistically non-significant shifts in baseline size distribution and CCN potential activation fraction; 2) Polluted continental and marine mixture - including January, April, May, and October 2017 events originated in continental industrialized areas, which caused statistically significant changes in baseline submicron particle number concentration, baseline size distribution, and baseline CCN potential activation fraction, 3) Biomass burning - including August and September 2017 events, which caused statistically non-significant changes in baseline submicron aerosol particles, but did produce statistically significant shifts baseline in particle size distribution and an increase in the CCN potential activation fraction.
> In the following three sections, we discuss case studies representatives of the diverse continental plumes arriving at ENA through the year, while in section 3.2.4, Table 3, and Fig. 8, we provide a summary statistic of the three multiday event regimes mentioned above."

> **Page 15, Line 4**, "Here, we provide a summary statistic of the influence of continental aerosol emissions on baseline aerosol population and baseline CCN concentrations at ENA for the three multiday event regimes discussed in Section 3.2."

**4.[Referee #1]**: *Given the diversity of the origins of these events, why is it appropriate to summarize the results of all of them together? (p.2 line 5) It would seem that averaging such events dampens the differences between them rather than showing how they contribute to variability.*

**[Resp.]:** We agree with Referee #1 that the total $N_{CCN}$ increase value due to the sum of all the events occurring in the year 2017 does not introduce any new valuable findings and might lead to dampening the different influences of transported aerosol transport event regimes on $N_{CCN}$ variability at ENA. We amended the text in the revised track-changes version of manuscript as follow:

> **Page 2, Line 4**: "Based on our analysis, in 2017, the multiday aerosol plume transport events dominated by mixture of dust and marine aerosol, mixture of marine and polluted continental aerosols, and biomass burning aerosols caused increases in $N_{CCN}$ baseline regime of respectively 6.6%, 8%, and 7.4% at SS 0.1% (and respectively 6.5%, 8.2%, and 7.3% at SS 0.2%) at ENA."

**5.[Referee #1]**: *Also the authors cite Wang et al. 2021, but I think a more quantitative and specific comparison to that work is needed to clearly show how this work improves/extends their results.*

**[Resp.]:** In order to address this concern, we have clarified the importance of our work within the overarching goals of the ACE-ENA field campaigns and its scientific objectives still to be addressed, in the Introduction. The new paragraph is reported below and shown in the track changes revised version of the manuscript.

> **Page 2, Line 15**: "The recent results from the ACE-ENA campaigns have advanced the knowledge of aerosol process (Zawadowicz et al., 2020; Wang et al., 2021c; Zheng et al., 2021), and cloud structures and processes (Gao et al., 2020; Yeom et al., 2021) in the remote MBL, as well as have allowed the evaluation of algorithms for remote sensing retrievals (Wu et al., 2020). However, many mechanisms underlying aerosol-cloud interactions over the North Atlantic remain unresolved. Within the ACE-ENA scientific objectives yet to be addressed, the complete understanding of the key controlling processes that shape CCN budget in the MBL is critical (Wang et al., 2021a). Motivated by this need, in this study, we leverage the AOS datasets collected at ENA during the entire year 2017 to constrain the influence of long-range transported particles with different continental origins on the cloud condensation nuclei concentrations in the pristine marine environment."

Moreover, through the text, we have expanded the comparison of our results to Wang et al., 2021 and to further literature focused on ENA, including recent studies outcomes of the ACE-ENA campaign, such as Rémillard and Tselioudis, 2015; Wood et al., 2015; Aiken et al., 2019; Giangrande et al., 2019; Zawadowicz et al., 2020; Sanchez et al., 2022. The additional comparison are shown in the track changes revised version of the manuscript as follow:

[revised manuscript text omitted]

*Specific Comments:*

**6.[Referee #1]**: *Table 3 Which events are summarized in each category? Need to specify here or in Table2. I think Table 3 would also benefit from some punctuation. It looks more like a ppt slide than an archival table.*

**[Resp.]:** We thank Anonymous Referee #1 for this comment. Table 3 has been revised and improved as per suggestion.

Table 3. Summary of the characteristics of each type of multiday aerosol plume transport event. Underlined values indicate statistically significant Δ

|  | **Dust and Marine mixture** | **Polluted continental and Marine mixture** | **Biomass Burning** |
|---|---|---|---|
| **Events Date (year 2017) and Origin** | • March 12 to 15 – Arctic/Canada
• November 26 to28 – North Africa
• December 07 to 10 – North Africa | • January 07 to 11 – North Europe
• April 20 to 22 – North Europe
• May 21 to 22 – North America
• October 11 to 13 – Hurricane | • August 26 to 29 – North America
• September 09 to 13 – North America |
| **Statistical analysis** | | | |
| $\Delta N_{tot}$ | > 110% | Between 95% and 110% | < 25% |
| $\Delta N_{At}/ N_{Ac}$ | < 1% | > 200% | > 200% |
| $\Delta AF_{0.1\%}$ | ~ 5% SS 0.1% | Between 30% and 75% | > 75% |
| $\Delta AF_{0.1\%}$ | ~ 7% SS 0.2% | Between 35% and 100% | > 75% |
| **Size mode fraction** | | | |
| $N_{At}$ contribution to $N_{tot}$ | ~ 59% | ~ 42% | ~ 33% |
| $N_{Ac}$ contribution to $N_{tot}$ | ~ 38% | ~ 56% | ~ 63% |

**7.[Referee #1]**: *Many places – significant increases of WHAT with respect to WHAT? (The latter is often missing.)*

**[Resp.]:** We have changed the text in the revised version of the manuscript to clarify that "significant increase/decrase" is refered to statistically significant changes in specific aerosol properties during the long-range transported aerosol plume events with respect to baseline conditions.

[revised manuscript text omitted]

**8.[Referee #1]**: *p.13 l.28 Why was Ntot higher for marine and dust? Seems like marine should be same or lower than baseline and dust would have low N (high M), so please explain.*

**[Resp.]:** We assume that Referee #1's comment is referring to the the first category of long-range transported continental aerosol events identified in the manuscript and called "mixture of dust and marine aerosol". When evaluating the changes induced by the arrival of these plumes on the baseline aerosol regime at ENA, we did not separate the relative influence due to marine aerosol from the relative influence due to dust. Instead we considered the changes that the whole mixture of dust and marine aerosols causes on baseline aerosol regime without differentiating between the two componenets. However, we aknowlodege that the statistically significant increase in baseline $N_{tot}$ during the event is mainly related to the presence of the dust component. In order to clarify this point, we changed the text in the revised version of the manuscript as follow:

**Page 1, Line 37,** "The arrival of plumes dominated by the mixture of dust and marine aerosol at ENA in the winter caused significant increases in baseline $N_{tot}$."

**Page 9, Line 42**, "Similarly, air masses from Arctic might also represent a source of mineral dust at ENA in the spring (Zheng et al., 2018). In the case of the Arctic, the atmospheric load of dust particles is the result bare soil surface and glacial outwash plains and it is projected to increase over the next years as consequence of the retreat of glaciers (Bullard et al., 2016; Tobo et al., 2019). In accordance with these observations, we found two events of southward transport from northern African and Portuguese flows to ENA in the months of November and December 2017, likely favoured by Arctic anticyclone, polar vortex, and midlatitude circulation, and an event of transport from Arctic in March 2017."

**Page 11, Line 2**, "During the transport over the ocean, dust particles typically mix with marine aerosols (Peshev et al., 2019) undergoing heterogenous chemical reactions and removal mechanisms that alter their composition and size and as a consequence their influence on the CCN aerosol baseline regime."

**9.[Referee #1]**: *How was baseline defined?.*

**[Resp.]:** This information is included in Section 3. We have changed the text to further clarify this point in **Page 7, Line 4** of the revised manuscript as:

"Here, we apply the algorithm to detect multiday transported aerosol events at ENA during the year 2017. Measurements affected by local aerosol events were removed prior the application of the algorithm following Gallo et al. (2020). Once the events have been identified, we removed the measurements affected by the arrival of continental aerosol plumes and we extract the aerosol baseline conditions (period of times not affected by local aerosol events and/or long-range transported plumes) to assess the aerosol seasonal regimes at ENA (Section 3.1)."

**Section 2.2**

**10.[Referee #1]**: *Averaging aerosol properties for 6 hour periods results in a coarse time resolution. A recent study by Dedrick et al. 2022 using ARM instrument aerosol properties to define marine and non-marine periods in the Southeast Atlantic shows moderate variability with 2 hour averaging periods. Please state the reason for 6-hr averaging.*

**[Resp.]:** We developed the algorithm to detect multiday transported aerosol plume events by reviewing methods utilized in previous works conducted in marine regions, and subsequently by testing different parametrizations including the utilization of mean and median values calculated over different periods of time. The reasons why we decided to use 6- hour periods are the following: 1) we aimed to identify only major continental plume events able to substantially influence the seasonal MBL CCN regime at ENA, while short duration events were not considered, 2) previous literature has reported that 6 h periods are sufficiently short to detect variations in air mass properties and at the same time long enough to mask the unwanted effect of diurnal cycle and processes that occur on small timescale (Wood at al., 2017, Dadashazar et al. 2021), 3) 6 hours periods match the time frequency of the Hysplit back-trajectories. We thank Referee #1 for pointing out that the reasons of our choice are not clearly stated in the manuscript

and we agree that without further explanations it sounds arbitrary. We have clarified the explanation on **Page 5, Line 30** of the revised manuscript:

> "The measurements are averaged over 6-hour periods which are sufficiently short to detect variations in mass properties but also sufficiently long to remove the effect of hourly variations due to diurnal cycles and process that occur on small timescale (Wood et al., 2017; Dadashazar et al., 2021) and match the time frequency of the Hysplit backward trajectories discussed below (the utilization of 7- and 8-hour periods was also tested and lead to the same results). The thresholds […]"

**11. [Referee #1]**: *How high is the variability of your aerosol properties using 6 hour averaging periods?*

**[Resp.]:** Variabilities between multiple 6-hours periods within the same detected events are up to 1.6-fold for median number concentration of $D_p$ 100 - 1000 nm particles, up to 1.3-fold for median submicron single scattering albedo at $\lambda$ 464 nm < 0.95, and up to 2.9-fold for mean black carbon concentrations > 40 ng m$^{-3}$.

**12. [Referee #1]**: *There is a lot of comparison/citation to Mace Head, but is that really the most appropriate comparison for ENA? Please consider a more broad consideration of the literature for some discussions, and/or please justify why Mace Head is same.*

**[Resp.]:** We assume that Referee #1's comment is referring to our choice of the black carbon concentration threshold > 40 ng m$^{-3}$ for the aerosol plume events identification. Although we cited a number of works conducted at Mace Head, we did base our determination of a BC threshold at ENA on further literature focused on different location across the Atlantic Ocean, namely:

- Saliba et al. 2020, which used BC threshold of 50 ng m$^{-3}$ to separate marine and continental periods in the Western North Atlantic during the NASA NAAMES field campaigns (and not at Mace Head as erroneously reported in the manuscript and pointed out by Referee #1 in the technical corrections section);
- Lawler et al. 2020, which characterized as marine ambient aerosol samples with BC < 50 ng m$^{-3}$ during the NASA NAAMES field campaigns in the Western North Atlantic – not cited.;
- Quinn et al. 2019, which reported BC concentrations between 15 and 25 ng m$^{-3}$ under clean marine conditions in the Western North Atlantic – not cited.
- Sakerin et al. 2021, which reported average BC concentrations between 37 and 44 ng m$^{-3}$ during cruise expeditions conducted between 2007 and 2020 over North Atlantic ocean – not cited.
- Pohl et al. 2014, which used as clean air background concentration of BC ranges from 20 and 44 ng m$^{-3}$ in the subtropical Atlantic.

We believe that expanded body of literature considered allowed us to provide an accurate estimation of BC concentrations at ENA during period affected by major plume events. We acknowledge that the text is not clear and does not reflect properly the research we have conducted. To clarify this point, we have revised section 2.2 and added additional references as following (**Page 5, Line 39** of the revised manuscript):

> "Black carbon concentrations ranging between 10 and 40 ng m$^{-3}$ during clean conditions have been reported by field studies conducted in different locations in the North Atlantic (O'Dowd et al., 2004; Shank et al., 2012;; Cavalli et al., 2016). Quinn et al. (2019) and Sakerin et al. (2021) have reported average BC concentrations between 15 and 25 ng m$^{-3}$ and 37 and 44 ng m$^{-3}$ respectively in the Western North Atlantic during the NAAMES field campaigns and during cruise expeditions conducted between 2007 and 2020 over North Atlantic ocean. A

threshold of 75 ng m$^{-3}$ has been typically utilized to indicate the presence of continental influenced air masses (Cooke et al., 1997; Kleefeld, 2002; Junker et al., 2006), while Pohl et al. 2014 have been used BC concentrations ranging from 20 and 44 ng m$^{-3}$ to identify clean background in the subtropical Atlantic. In more recent works, Facchini et al. (2008) and O'Dowd et al. (2014), determined BC 50 ng m$^{-3}$ as a threshold value to identify combustion influences at Mace Head. Similarly, Saliba et al. (2020) and Lawler et al. (2020) used the same criterion to separate ambient marine from continental periods in the Western North Atlantic."

**13.[Referee #1]**: *What is a phytoplankton-derived aerosol? Do you mean sulfate from DMS?*

**[Resp.]:** Wording has been revised and changed to "aerosols generated by phytoplankton activities at the surface ocean" **Page 9, Line 20** of the revised manuscript.

**14.[Referee #1]**: *Does it result in a different amount of multi day aerosol plume transport events?*

**[Resp.]:** We assume that Referee #1's comment is referring to the the utilization of different averaging periods for the identification of the events. In our study we tested two additional averaging time periods of 7- and 8-hours. The events identified using the averaging time periods of 7- and 8-hours where the same we detected using 6-hours averaging time periods. Therefore, based on these results and on on the reasons reported in our response to Referee #1's comment #10 (above), we found the choice of using 6 hours averaging time periods to be the most appropriate. This point has been clarified on **Page 5, Line 30** of the revised version of the manuscript:

> "The measurements are averaged over 6-hour periods which are sufficiently short to detect variations in mass properties but also sufficiently long to remove the effect of hourly variations due to diurnal cycles and process that occur on small timescale (Wood et al., 2017; Dadashazar et al., 2021) and match the time frequency of the Hysplit backward trajectories discussed below (the utilization of 7- and 8-hour periods was also tested and lead to the same results). The thresholds […]"

**15.[Referee #1]**: *Why do certain aerosol properties use mean or median to define thresholds?*

**[Resp.]:** We developed the algorithm upon an extended investigation and comparison to previous works describing aerosol properties under clean background conditions versus continental periods at ENA and in locations comparable to ENA, aiming to identify the most appropriate parameters and respective thresholds to ingest in the algorithm. Within these studies, Pennypacker and Wood (2017) utilized median number concentration of particles with $D_p$ 100-1000 nm and SSA at ENA to minimize the potential impacts of short-duration unidentified outlier data which would likely bias the mean values. Simulatenously, mean has been widely utilized in previous literature for reporting BC levels in different locations of the North Atlantic. In our study we removed periods impacted by local aerosol events prior conducting any data analysis and we obtained similar mean and median values for three paramenters considered. A comparison of mean and median values for number concentration of particles with $D_p$ 100-1000 nm, SSA, and BC for the three events described in the manuscript is shown in the table below. Based on these results, we chose to use median values for number concentration of particles with $D_p$ 100-1000 nm and SSA measurments, and mean values for BC concentrations to allow a better comparison of our results to the previous literatures reported in Section 2.2.

| Event | Number concentration particles $D_p$100-1000 nm | | Median SSA 1 µm (λ 464 nm) | | Mean BC (ng m⁻³) | |
|---|---|---|---|---|---|---|
| | Mean | Median | Mean | Median | Mean | Median |
| April 20 to 22 | $467 \pm 65$ cm⁻³ | 460 cm⁻³ | $0.941 \pm 0.03$ | 0.94 | $121 \pm 27$ ng m⁻³ | 113 ng m⁻³ |
| September 09 to 13 | $301 \pm 78$ cm⁻³ | 289 cm⁻³ | $0.939 \pm 0.01$ | 0.93 | $175 \pm 39$ ng m⁻³ | 180 ng m⁻³ |
| December 07 to 10 | $236 \pm 64$ cm⁻³ | 235 cm⁻³ | $0.918 \pm 0.02$ | 0.92 | $103 \pm 18$ ng m⁻³ | 97 ng m⁻³ |

We thank Referee #1 for pointing out that the reason of our choice is not clearly stated in the manuscript. We have ameneded the text as follow to clarify the explanation on **Page 6, Line 14** of the revised track-change manuscript:

> "It is important noting that the utilization of medians instead of means for number concentration of $D_p$ 100 - 1000 nm particles and SSA to constrain periods impacted by long-range transport events in Pennypacker and Wood (2017) is due to the need of minimize the potential impacts of unidentified outlier. In our study, we performed post data processing methods prior conducting any data analysis to removed short-duration high concentration aerosol events (Gallo et al., 2020) and we obtained similar mean and median values (difference between mean and median values < 12%) for the three parameters used to develop the multiday transported aerosol plume event identification algorithm. Therefore, to allow a better comparison of our results to the previous literatures, the algorithm relies on the utilization of median values for number concentrations of particles with $D_p$ between 100 and 1000 nm and submicron SSA at at 464 nm wavelength, and mean values for the black carbon concentration."

**Section 3.1.1**

**16.[Referee #1]**: *Entrainment is mentioned several times in this paper. Have you looked into separating aerosol property data using proxies for entrainment rate such as delta-T at top of MBL?*

**[Resp.]:** The influence of entrainment rate on aerosol properties has not been investigated, and although certainly being of considerable interest, is beyond the scope of this study. However, in the manuscript we provide an extensive comparison of our results to previous work (such as Zheng et al., 2018) focused on the quantitative understanding of aerosol key controlling processes (including entrainment from the free troposphere) over the North Atlantic ocean, (see the example reported below and **Section 3**). As such and as acknowledge by Referee #1 in the general comments, we feel that our findings are strongly supported by past literature.

> **Page 7, Line 33**, "in the summer, MBL baseline aerosol concentrations might be influenced by the entrainment of diluted and aged continental particles from the free troposphere which likely contributes to enhanced concentration of particles in the Accumulation mode (Wang et al., 2021a). This observation is consistent with previous studies investigating aerosol vertical profiles during the summer ACE-ENA field campaign (Wang et al., 2021b), and over the Western North Atlantic during the NASA North Atlantic Aerosol and Marine Ecosystems Study campaign (NAAMES)."

**Section 3.1.1 and Section 3.1.2**

**17.[Referee #1]**: *The paper discusses the removal of large particles by precipitation several times. What happens when you separate the aerosol properties that follow precipitation events?*

**[Resp.]:** Aerosol properties as a function of precipitation events were not assessed in this work. The characterization of precipitation events needed for accurately determining how precipitation influences aerosol properties, would require a detailed analysis of the cloud regime precipitation and precipitation domain at ENA, which is beyond the scope of this study. Instead, we have utilized a number of previous works and well established assumptions to link the observed seasonal variations in aerosol concentration and size, $N_{CCN}$, and potential activations fraction to emission sources and atmospheric processes such as in-cloud precipitation removal. In the revised version of the manuscript, we have improved the discussion of Sections 3.1.1 and 3.2.2 by expanding the comparison of our findings to previous works conducted at ENA.:

> **Page 9, Line 14,** "Supporting our finding, Wang et al. (2021a) reported high precipitation rate and increase CCN coalescence scavenging, accompanied by enhanced $N_{Ac}$ activation at ENA during the ACE-ENA winter field campaign."

> **Page 9, Line 21**, "The higher summertime $N_{CCN}$ observed here are in agreement with previous studies conducted at ENA which also found a correlation between elevated $N_{CCN}$ and concentration of cloud droplet (Wood et al., 2015; Wang et al., 2021a) and reduced precipitation (Rémillard and Tselioudis, 2015; Giangrande et al., 2019), thus suggesting minimal CCN removal through wet scavenging. Simultaneously, strong VOC emissions at the surface ocean due to the final phase of the phytoplankton bloom and microbial activities leads to the formation of highly hygroscopic secondary sulfate particles which grow quickly into CCN by condensation and well explain the elevated $N_{CCN}$ and potential activation fractions found here (Saliba et al., 2020; Zawadowicz et al., 2020)."

**18.[Referee #1]**: *The paper also discusses the effect of wind speed on large Ac mode several times. How well do wind speed and parameters of the large Ac mode such as mean diameter and number concentration correlation at the ENA?*

**[Resp.]:** Zheng et al., 2018 assessed the correlation between wind speed and particle concentration and size at ENA over a 3-years period including the year 2017 focus of our study. They reported no correlation between the concentration of particles in the Ac mode and wind speeds. We have added the following text in the revised manuscript (**Page 8, Line 19**) to support our findings and improve the comparison of our work to previous literature.

> "On the contrary, sea spray aerosol production at the surface ocean due to enhanced winter-time wind speeds up to 21.7 m s$^{-1}$ (Aiken et al., 2019) substantially contributes to Large Accumulation modes concentrations explaining the higher $N_{La}$ observed in January (Vignati et al., 2010; Zheng et al., 2018; Quinn et al., 2019b). During late spring and summer, the phytoplankton bloom is responsible for strong ocean emissions of dimethylsulfide, whose oxidation products have been found to enhance the condensational growth of nucleation mode particles into the Aitken and subsequently to the Ac modes (O'Dowd et al., 1997; Andreae et al., 2003; Zheng et al., 2018). Furthermore, photochemistry and/or oxidation of oxygenated gas-phase organic compounds of marine origin produce secondary organic aerosols at the surface layer which contribute to the growth of Aitken mode particles during late summer when phytoplankton activity is lower (Mungall et al., 2017). In a previous study conducted at ENA between 2015 and 2018, Zheng et al. 2018 assessed the correlations between wind speeds and particle size. In the summer, no correlations between wind speeds and $N_{At}$, and $N_{Ac}$ were

reported while $N_{LA}$ was observed to strongly correlate with wind speeds, therefore suggesting that the contribution from sea spray is limited to the Large Accumulation mode."

**Section 3.2.4**

**19.[Referee #1]**: *The paper introduces HSD to define statistically significant changes on baseline aerosol number concentrations, aerosol size modes, and CCN potential activation fraction. However, scattered throughout the paper in sections before that significance is also used interchangeably to describe differences in seasonal statistics and baseline conditions. I recommend a different word or plainly writing out the quantitative differences to avoid confusion.*

**[Resp.]:** We thank Referee #1 for pointing out this issue. To improve clarity we screened the manuscript and: 1) substituted the words "significance" and "significantly" with synonymous if not refereed to statistically significant difference, 2) added the word "statistically" to significant anytime the latter was referend HSD statistically significant variations.

**20.[Referee #1]**: *In this paper, the authors use activated CCN fraction and N_tot to speculate whether aerosol composition or increased aerosol concentration affect CCN at the ENA. Have collocated cloud properties (by either ARM ground or NASA satellite retrievals) such as cloud effective radius been analyzed for these case studies? It would be convincing to see if there is a statistically significant difference in cloud properties versus baseline conditions due to rapid cloud adjustments.*

**[Resp.]:** The main focus of our work is investigating the effect of aerosol perturbations on CCN regime at ENA, while the influences on cloud properties and cloud adjustment mechanisms have not been analyzed. Furthermore, although ENA is equipped with three state-of-the-art second-generation radar systems, in a recent study Lamar et al. (2019) reported variability between instruments when analyzing short periods of time, therefore challenging the ability of characterizing rapid cloud adjustments. Recently, ARM has released a new value-added product (MFRSRCLDOD) that provides retrieved cloud optical properties at ENA. We agree with Referee #1 recogizing the importance of this topic. Therefore, the utilization of this ARM product as well as ARM ceilometer lidar and KAZR2 datasets might be explored in the future to evaluate the influence of aerosol perturbations on clouds properties over ENA, upon evaluations of radar-lidar techniques and validation of retrieved observations against in situ measurements. We added this comment in the Conclusion section on **Page 17, Line 7** of the revised manuscript:

> "Furthermore, the influences of aerosol perturbations on cloud properties and cloud adjustment at ENA might be explore in future studies using ARM retrieved cloud optical properties value-added products as well as ARM ceilometer lidar and KAZR2 datasets upon evaluations of radar-lidar techniques and validation of retrieved observations against in situ measurements."

**Technical Corrections**

**21.[Referee #1]**: *There are a significant number of typos. Some are noted below. Please reread and check for these.*

**[Resp.]:** We thank Referee #1 for careful reading our manuscript and noticing the typos. We have amended the revised manuscript correcting the typos and clarifying the wording when needed. All the alterations are shown in the track changes revised version of the manuscript and in our point-by-point responses below.

- *p.6 line 8 "era"*
  Corrected to "are".

- *p.5 line 21 Saliba et al. was not at Mace Head*
  Changed to: "Saliba et al. (2020) and Lawler et al., (2020) used the same criterion to separate ambient marine from continental periods in the Western North Atlantic."

- *p.1 line 34 "mixture of dust and marine aerosols from North Af" – is the marine from N.A. too or is that just the dust?*
  Corrected to "mixture of marine aerosols and dust from North Africa".

- *p.2 line 1 cloud concentration nuclei concentrations*
  Corrected to "Consequentially, cloud nuclei concentrations increased ~115%".

- *Overall, please stay consistent with usage At mode and Ac mode versus fully writing out Accumulation Mode and Aitken mode.*
  Corrected in the revised version of the manuscript.

- *Section 2.1 "is given" should be "are given"*
  The subject of the sentence is "a list" which is singular, therefore we believed the verb has to be "is" ("A list of the AOS measurements analyzed here, including references for each instrument, is given in Table 1 and summarized in the following sections.").

- *Section 3.1.1 Please define what months belong to which seasons earlier in the paper as you discuss summer mean values before doing so.*
  Winter referes to the month of winter January – February, and November -December 2017, while summer refers to the months of June to Septemebr 2017. The information has been added in the text and showed below:

  "[…]submicron particles in the winter (Jan. – Feb., and Nov. -Dec. 2017) and higher […]"
  "[…] we observed that summer (June – September 2017) mean $N_{Ac}$ values, which […]"

- *Section 3.1.1 "The influence of local aerosol sources on Ac mode aerosols measurements at ENA is minimal" can be written more concisely as "There is minimal influence of local aerosol sources on Ac mode aerosol measurements at ENA"*
  Corrected to "There is minimal influence of local aerosol sources on Accumulation mode aerosols measurements at ENA".

- *Section 3.2 Line 40 Can remove "specific" in front of case studies to reduce redundancy*
  Corrected to "[…] we discuss case studies representatives of the diverse continental plumes […]".

- *Section 3.2.3 Line 13 Missing space before "Here we"*

*Corrected.*

- *Section 3.2.3 Line 14 Missing space after "September 09th". "September 09th" should also be "September 9th". "During the period in analysis," can be written more concisely as "During this period". Section 3.2.3 Line 23 Can remove "under" to be more concise.*
Corrected to "[…] affected ENA between September 9th and September 13th, 2017. During this period, the arrival of air masses from North America are associated with […]".

- *Section 3.2.3 Line 32 This sentence is worded confusingly.*
Reworded as: "Associated to the above-mentioned shift in size distribution, we found potential activation fractions (0.44 and 0.70 at SS 0.1% and 70% at SS at 0.2%, respectively) approximately twice that under baseline conditions, suggesting that the transported aerosol particles are more effective as CCN."

- *Section 3.2.3 Line 37 Add comparison values in parenthesis to your percentage increases.*
Corrected to "The CCN concentration was higher (respectively 220% at SS 0.1%, and 227% at SS 0.2%) during the event then for rest of the month of September 2017.".

- *Section 3.2.3 Line 37 Can more concisely word this as "aged wildfire aerosols dominate the accumulation mode and act better as CCN"*
Corrected to "These results demonstrate that aged wildfire aerosols dominate the accumulation mode and act better as CCN".

- *Section 4 Line 23-24 add "a" before "mixture" and remove "," after "March 2017"*
Corrected to "a mixture of dust and marine aerosols from the Arctic and Canada in March 2017 and from North Africa in November and December 2017,".

- *Section 4 Line 25 add "a" before mixture*
Corrected. To "a mixture".

- *Section 4 Line 26 "form" should be "from"*
Corrected to "from".

- *Section 4 Line 27 "the aerosol plumes composition" can be written as "aerosol plume composition"*
Corrected to "the influence of aerosol plumes composition".

- *Section 4 Line 29 "causeed" should be "caused"*
Corrected to "caused".

- *Section 4 Line 30 " Mixture of marine and polluted continental aerosol plumes showed high CCN concentrations attributable to both high Ntot, and predominance of large particles (Dp > 100 nm) of sufficient size to readily serve as CCN." can be written in active voice and more concisely as "High CCN concentrations are attributed to both high Ntot and dominance of particles large enough to act as CCN (Dp > 100 nm) from mixed marine and polluted continental aerosol plumes."*
Corrected to "High CCN concentrations are attributed to both high $N_{tot}$, and dominance of particles large enough to act as CCN ($D_p$ > 100 nm) from mixed marine and polluted continental aerosol plumes.".

- *Section 4 Line 35 Move ",in 2017," to the end of the sentence.*

"in 2017" moved to the end of the sentence.

**Reference**

[revised manuscript text omitted]
_{4r}$ / $N_{4c}$) $N_{4r}$ / $N_{4c}$ change < 1% $N_{4r}$ contribution to $N_{tot}$ ~ 59% $N_{4c}$ contribution to $N_{tot}$ ~ 38% | Statistically significant shift in size ($N_{4r}$ / $N_{4c}$) $N_{4r}$ / $N_{4c}$ change > 200% $N_{4r}$ contribution to $N_{tot}$ ~ 42% $N_{4c}$ contribution to $N_{tot}$ ~ 56% | Statistically significant shift in size ($N_{4r}$ / $N_{4c}$) $N_{4r}$ / $N_{4c}$ change > 200% $N_{4r}$ contribution to $N_{tot}$ ~ 33% $N_{4c}$ contribution to $N_{tot}$ ~ 63% |
| Statistically non significant change in CCN potential Activation fraction $AF_{0.1\%}$ increase ~ 5%, $AF_{0.2\%}$ increase ~ 9% | Statistically significant change in CCN potential activation fraction $AF_{0.1\%}$ increase between 30% and 75% $AF_{0.2\%}$ increase between 35% and 100% | Statistically significant change in CCN potential activation fraction $AF_{0.1\%}$ and $AF_{0.2\%}$ increase > 75% |

Table 3. Summary of the characteristics of each type of multiday aerosol plume transport event. Underlined values indicate statistically significant Δ

| | **Dust and Marine mixture** | **Polluted continental and Marine mixture** | **Biomass Burning** |
|---|---|---|---|
| **Events Date (year 2017) and Origin** | • March 12 to 15 – Arctic/Canada
• November 26 to28 – North Africa
• December 07 to 10 – North Africa | • January 07 to 11 – North Europe
• April 20 to 22 – North Europe
• May 21 to 22 – North America
• October 11 to 13 – Hurricane | • August 26 to 29 – North America
• September 09 to 13 – North America |
| **Statistical analysis** | | | |
| $\Delta N_{tot}$ | $\geq 110\%$ | Between 95% and 110% | < 25% |
| $\Delta N_{At}/ N_{Ac}$ | < 1% | $\geq 200\%$ | $\geq 200\%$ |
| $\Delta AF_{0.1\%}$ | ~ 5% SS 0.1% | Between 30% and 75% | $\geq 75\%$ |
| $\Delta AF_{0.1\%}$ | ~ 7% SS 0.2% | Between 35% and 100% | $\geq 75\%$ |
| **Size mode fraction** | | | |
| $N_{At}$ contribution to $N_{tot}$ | ~ 59% | ~ 42% | ~ 33% |
| $N_{Ac}$ contribution to $N_{tot}$ | ~ 38% | ~ 56% | ~ 63% |

[Figure]

**Figure 1**. Box and whisker plot of monthly ubmicron aerosol number concentrations (box bottom at 25%, box top at 75%, whisker bottom at 10%, and whisker top at 90%). Mean (circles) and median (open circles) of total number concentration (black), number of Aitken (yellow), and  Accumulation (green) modes.

[Figure]

**Figure 2**. Particle size distribution in winter (a) and summer (b) 2017, per each size bean mean circle, and median -, box bottom at 25%, box top at 75%, whisker bottom at 10%, and whisker top at 90%). Discontinuity at around 270 nm due to technical limitations of the UHSAS (handoff region between two internal gain stages).

[Figure]

**Figure 3.** Box and whisker plot of $N_{CCN,0.1\%}$ (a) and $N_{CCN,0.2\%}$ (b), mean $N_{CCN}$ blue circles, median -, box bottom at 25%, box top at 75%, whisker bottom at 10%, and whisker top at 90%, mean $N_{tot}$, black circles, and CCN potential activation fraction red circles.

**Figure 4**. CALIPSO trajectories (blue) and Hysplit back trajectories (red) arriving at 50 m a.g.l. above the ENA site on December 07, 2017 (© Google Earth 2015).

[Figure]

**Figure 5.** CALIPSO trajectories (blue), and Hysplit back trajectories (red) arriving at 50 m a.g.l. above the ENA site on April 21, 2017 (© Google Earth 2015).

[Figure]

**Figure 6.** NASA Worldview VIIRS 375 Active fires between September 1 and 15, 2017 (red circles), and Hysplit back trajectories arriving at 50 m a.g.l. above the ENA site on September 10, 2017 (© Google Earth 2015).

[Figure]

[Figure]

**Figure 7.** Case study of December 2017 (leftmost), April 2017 (center), and September 2017 (rightmost) events. Submicron particle size distribution under baseline conditions (blue) and during the events (red), and $\kappa_{HTDMA}$ (open circles) during the events (a, f, k), Aitken, Accumulation, and Large Accumulation mode contributions to (b, g, l), non-refractory sulfate and organic aerosols (c, h, m), scatter plot of $N_{CCN}$ versus $N_{tot}$ during the event (red circle) and fitting lines for the events at SS 0.1% (red) and at SS 0.2% (dark red) (d, i, n), plot of potential activation ratio versus $N_{Ac} / N_{At}$, or the events at SS 0.1% (red) and at SS 0.2% (dark red) (e, j, o).

[Figure]

**Figure 8**. Mean percentage change in $N_{tot}$, $N_{At}$, $N_{Ac}$, $N_{CCN-0.1\%}$, and CCN potential activation fraction at SS 0.1% for each type of event (a); Aitken, Accumulation and Large Accumulation particle modes relative contribution to $N_{tot}$, for baseline and each type of event.

---

## Author Comment (AC2)

**Manuscript No.**: acp-2022-637

**Title**: Long-range transported continental aerosol in the Eastern North Atlantic: three multiday event regimes influence cloud condensation nuclei

**Responses to Anonymous Referee #2**

*General Comments:*

**1.[Referee #2]**: *Gallo et al. studied the influence of aerosol transport events on routine aerosol properties at the ENA marine background site. Using the data collected in 2017, nine multi-day events were identified and grouped according to their air origin and aerosol physical/optical properties. These events had a profound influence on the cloud condensation nuclei properties by increasing their concentration. The manuscript is well written and most of the analysis are sound and solid. However, the manuscript reads more like an ACP measurement report as not much new is brought to the table. The authors could try making a stronger case e.g. by adding more statistics (maybe even further data/years) and/or by adding a more detailed discussion on the difference to previous studies and this data can be used (e.g. for model improvements or validation exercises). Furthermore, some important technical details are currently missing and should be added to the revised version. I recommend major revisions.*

**[Resp.]:** We thank Referee #2 for taking the time to review our manuscript and support our work. We have addressed Referee #2's concerns and improved the manuscript by furthering comparing our findings to previous literature in Introduction, Results and Discussion, and Conclusions Sections. We agree that expanding our analysis to multiple years would be of considerable interest to understand interannual variability at ENA, however the topic is beyond the scope of the current work. We think that the revised version of the manuscript has a stronger impact and brings new conclusions for improving the understanding of critical aerosol processes in the marine regions. As such, we believe that the revised manuscript fulfills the requirements needed to be published as an ACP Research article. Please find our itemized responses in below (blue) and the corrections/alterations in the track changes revised version of the manuscript. Note that throughout this response, the original Referee #2's comments are highlighted in italic black and our responses follow in blue.

*Detailed Comments:*

**2.[Referee #2]**: *Detailed comments are given below.*

**[Resp.]:** We thank Anonymous Referee #2 for raising these points. We have revised the manuscript per the suggestions. All detailed comments are addressed in the following point-by-point discussions below and the corrections are shown in the track changes revised version of the manuscript.

*Page 2, line 4-8: The last two sentences in the abstract should be revised since they are difficult to understand. Maybe add the percentages to the different transport types. What do you mean by the last sentence? How will this be possible without detailed knowledge of the chemical composition?*

**[Resp.]:** We thank Referee #2 for pointing out the lack of clarity in the Abstract. Chemical composition data have also been added in the Results and Discussion section to strength our findings. See our response to Reviewer #1's comment 4 that addresses this same topic. The last sentences of the Abstract in the revised version of the manuscript have been revised to read as:

> **Page 2, Line 4**: "Based on our analysis, in 2017, the multiday aerosol plume transport events dominated by mixture of dust and marine aerosol, mixture of marine and polluted continental aerosols, and biomass burning aerosols caused increases in $N_{CCN}$ baseline regime of respectively 6.6%, 8%, and 7.4% at SS 0.1% (and respectively 6.5%, 8.2%, and 7.3% at SS 0.2%) at ENA."

*Page 3, End of Sect 1: One way to improve the manuscript could be to specifically state the research questions here. What are you trying to find out? And how does this lead to an advancement? Why is it important?*

**[Resp.]:** We thank Referee #2 for this suggestion. We have added two sentences in the Introduction which state the overarching goals of our study as well as its importance within the objectives of the ACE-ENA campaign and the potential implications that the findings might have on improving climate models. The new text is reported below and shown in the track changes revised version of the manuscript.

> **Page 4, Line 3**: "With this study, we aim to provide key observational constraints to parametrize the influence of changes in baseline $N_{tot}$ and particle size modes due to aerosol perturbation events on CCN baseline regimes. Ultimately, our results might be used as a proxy to estimate the CCN budget over remote oceans and to inform climate model improvements and validation."

*Section 2: Although many studies have been published using the aerosol data from ENA, it is still needed to describe a few technical details on the sampling and your 2017 data:*

**[Resp.]:** We thank Referee #2 for these suggestions. Please note that in the manuscript we only briefly describe the ARM Aerosol Observed System (AOS) and the instruments within it. Detailed technical descriptions, including set up and operation of the instruments, technical specifications, and maintenance and calibration procedures, goes beyond the scope of this manuscript and are provided elsewhere as indicated in the manuscript. We have addressed Referee #2's concerns about the potential lack of these information by carefully review the references in the Measurements and methodology section and by adding new citations and amending the text when needed. Please, find the references that were added to the revised version of Sect. 2 and our point-by-point responses below.

Reference: Ng, N. L., Herndon, S. C., Trimborn, A., Canagaratna, M. R., Croteau, P. L., Onasch, T. B., Sueper, D., Worsnop, D. R., Zhang, Q., Sun, Y. L., and Jayne, J. T.: An Aerosol Chemical Speciation Monitor (ACSM) for Routine Monitoring of the Composition and Mass Concentrations of Ambient Aerosol, Aerosol Sci. Technol., 45, 780–794, https://doi.org/10.1080/02786826.2011.560211, 2011.
Springston, S.: Particle Soot Absorption Photometer (PSAP) Instrument Handbook, https://doi.org/10.2172/1246162, 2018;
Uin J., and Smith S.: Eastern North Atlantic (ENA) Aerosol Observing System (AOS) Instrument Handbook, 2020;
Uin, J.: Ultra-High-Sensitivity Aerosol Spectrometer (UHSAS) Instrument Handbook, https://doi.org/10.2172/1251410, 2016a;
Uin, J.: Cloud Condensation Nuclei Counter (CCN) Instrument Handbook, 2016b;

Uin, J.: 3002 Humidified Tandem Differential Mobility Analyzer Instrument Handbook, 2016c;

Uin, J.: Integrating Nephelometer Instrument Handbook, https://doi.org/10.2172/1246075, 2016d;

Watson, T. B.: Aerosol Chemical Speciation Monitor (ACSM) Instrument Handbook, https://doi.org/10.2172/1375336, 2017.

- *What kind of inlet was used (with or w/o size cut)?*
  The ARM Aerosol Observed System comprised of one container that samples aerosols using instrumentations connected to a central inlet located approximately 10 m above ground. Most of the instruments do not have a size cut, exceptions are the Nephelometer and PSAP. Detailed information on the AOS can be found in Bullard et al., (2017), and Uin et al., (2019) as indicated in the Section 2.1 of the manuscript. Reference about the Nephelometer (Uin, 2016) was also added. To clarify this point the text has been amended as follow:

  **Page 4, Line 11**: "The ENA ARM AOS comprises of one container that samples aerosols using instrumentations connected to a central no-heated inlet inlet located approximately 10 m above ground."

  **Page 5, Line 16**: "ARM archive Nephelometer data includes corrections for truncation and illumination errors. Prior to measurement, the PSAP and nephelometer sample air passes through an impactor that periodically switches between 1 and 10 μm cut-point sizes (Uin et al., 2019). $B_{abs}$ and $B_{sca}$ values discussed in this study refer to measruements collcted at 1 μm cut-point sizes."

- *Was the inlet heated?*
  No. No parts of the AOS sample system are deliberately heated. When drying is required, it is done using nafion dryers, as reported in Uin et al. (2019). The information has been added in the revised manuscript.

  **Page 4, Line 12**: "The ENA ARM AOS comprise of one container that samples aerosols using instrumentations connected to a central no-heated inlet inlet located approximately 10 m above ground."

- *What was the average RH before the aerosol instrumentation?*
  We assume that Referee #1's comment is referring to ambient RH at ENA. The mean RH value during the year is 74% ± 11%. The RH for the AOS stack is reported in Uin et al. 2019 and is < 40%.

- *What was the average RH for the dry diameter (first DMA in the HTDMA)?*
  The HTDMA average dry diameter RH ranges between 6.1% and 7.3% and typically lower than 10%. The information has been added in the revised version of the manuscript.

  **Page 4, Line 37**: "Particle hygroscopic growth (HG) at subsaturated conditions is calculated as the ratio of the geometric mean mobility diameter of the humidified particles ($d_m$(RH)) (RH > 85%) to the dry diameter ($d_d$) (RH between 6.1% and 7.3%)."

- *How and how often were the CCNC and HTDMA calibrated?*
  Typically, CCN counter and HTDMA instruments are calibrated once per year. ARM common procedures are reported in the ARM instument handbooks. Reference to the ARM AOS instrument handbooks have been added in **Table 1**.

**Table 1. Aerosol Observing System measurements** at ENA ARM site analyzed in this study.

| Measurement | Symbol | Unit | Instrument | Reference |
|---|---|---|---|---|
| Submicron aerosol number concentration | $N_{tot}$ | cm$^{-3}$ | Condensation Particle Counter CPC Model 3772, TSI Inc. | (Kuang et al., 2019) |
| Size distribution of submicron aerosols (70 to 1000 nm) | | cm$^{-3}$ | Ultra-High-Sensitivity Aerosol Spectometer UHSAS, DMT | (Uin et. al, 2016a) |
| Number concentration of cloud condensation nuclei | CCN | cm$^{-3}$ | Cloud Condensation Nuclei Counter CCN Model CCN-100, DMT | (Roberts and Nenes, 2005; Rose et al., 2008; Uin et. al, 2016b) |
| Aerosol growth factor | | | Humidified Tandem Differential Mobility Analyzer HTDMA Model 3002, Bretchel | (Lopez-Yglesias et al., 2014; Uin et. al, 2016c) |

| | | | | |
|---|---|---|---|---|
| Aerosol absorption coefficients | $B_{abs}$ | Mm$^{-1}$ | Particle Soot Absorption Photometer PSAP 3-λ, Radiant Research | (Bond et al., 1999; Virkkula et al., 2005; Virkkula, 2010; Springston, 2018) |
| Aerosol scattering coefficients | $B_{sca}$ | Mm$^{-1}$ | Integrating Nephelometer Neph, Model 3563, TSI | (Costabile et al., 2013; Uin et. al, 2016d) |
| Non-refractory sufate and organic | | μm$^{-3}$ | Aerosol Chemical Speciation Monitor Aerodyne Research | (Ng et al., 2011; Watson, 2017) |

- *Where the scattering coefficients corrected for truncation and illumination errors?*
  Yes, the information has been added in Sect. 2.1.2 and it can be found in Uin (2016d) which has been added to the reference.

    **Page 5, Line 16**: "ARM archive Nephelometer data includes corrections for truncation and illumination errors (Uin, 2016d)."

- *Are the values given at ambient pressure or corrected to STP?*
  We added the information to the revised version of the manuscript.

    **Page 4, Line 14**: "Pressure for aerosol instruments is given at ambient conditions if not differently stated."

- *How much data was removed and how complete is the entire 2017 dataset? Maybe add a table to the SI.*
  We interpret and respond to this comment as being in reference to the data points impacted by local aerosol events and removed using the ENA-Aerosol Mask algorithm (Gallo et al., 2020) prior to conducting the data analysis reported in this study. For the year 2017, ENA-AM removed ~23% of the data. The information was added in Sect. 2.1.

    **Page 4, Line 16**: "Prior to conducting any data analysis, periods impacted by local aerosol events (~23% of the 2017 datasets used in the study) were removed from submicron aerosol number concentration […]"

- *Page 4, line 8: I would remove the word "optical". Although the CPC detects optically the individual particles, the lower cut-off diameter is determined by the settings and technical details of the CPC (e.g. reached supersaturation).*
  The word "optical" gas been removed in the revised version of the manuscript.

- *Page 5, line 29: GDSA -> GDAS*
  Corrected to GDAS.

- *Page 7, line 18: Since the phytoplankton activity is low, are the oxygenated gas-phaseorganic compounds of marine or transported origin?*
  This is referred to compounds of marine origin. We clarified this point in the revised version of the manuscript.

    **Page 8, Line 27**: " Furthermore, photochemistry and/or oxidation of oxygenated gas-phase organic compounds of marine origin produce secondary organic aerosols at the surface layer which contribute to the growth of Aitken mode particles

- *Page 7, line 35: The last value should be at 0.2% SS, correct?*
  Yes, Referee #2 is correct. The value has been corrected in the revised version of the manuscript.

- *Page 7, line 37: Strictly spoken you did not observe a "reduced biological activity" but rather lower number concentration. Suggest to re-phrase this sentence.*
  The sentence has been rewritten in the revised version of the manuscript and it reads as:

> **Page 9, Line 9**: " The low number particle concentration and consequently low concentrations of cloud condensation nuclei observed in the MBL can be to a large degree attributable to reduced ocean biological activity in the winter."

- *Page 8, line 21 (and also later in the text and table): Artic -> Arctic*
  Corrected to Arctic.

- *Page 9, line 9: Suggest to round all kappa-values to 2 digits after the comma.*
  Corrected.

- *Page 12, line 6: Please add a reference or toolkit used for Tukey-Kramer test.*
  The information has been added in Sect. 2.2.

- *Page 12, line 17: Besides gravitational settling, it is probably also due to wet scavenging that coarse mode particles are removed.*
  Thank you for pointing out this hypothesis. We have added it in the revised version of the manuscript.

  > **Page 15, Line 20**: "Although wet scavenging might also played a role in the removal of coarse particles."

- *Table 2: Why aren't the kappa-values included here? This would be very useful s well.*
  Thank you for the suggestion. As indicated in the legend, Table 2 only reports the values of the three aerosol properties used by the algorithm to detect the events (median concentration of particles with $D_p$ 100-1000 nm, median SSA 1 μm at λ 464 nm, and mean BC values), therefore we feel the kappa-values (for only three cases) do not fit in in Table 2. However, kappa-values are shown in Fig. 7a, 7f, and 7h.

- *Table 3: burining -> burning*
  Corrected.

- *Sect. 4, last sentence: Why is that algorithm needed (if the site continuously measured CCN concentrations)? The reader is left a bit alone on why this is needed and could be important.*
  We thank Referee #2 for pointing out the lack of clarity in the sentence. We have amended the text as follow:

  > **Page 17, Line 3**: " Based on the characteristics of the type events discussed above, in the future an algorithm to predict $N_{CCN}$ variations during multiday events of long-range transport of aerosols could be developed and validateed at ENA to inform study at other locations and constrain model predictions of CCN regime perturbations over remote oceans."

- *The conclusions and abstract read both more like a summary and could be shortened.*
  We thank Referee #2 for raising this criticism. In the Abstract section, we do aim to provide a brief introduction of the topic followed by a summary that recapitulates the key points and findings of the study, as recommended in the ACP Manuscript composition guidelines. We also feel that the Conclusions section is important and relevant to summarize and highlight our main findings. In the Conclusion section, we have re-phrased the sentences that were lacking clarity and provided additional context for leveraging the potential of our study in informing future researches, as also suggested by Referee #2 in the previous comments (see answer to General Comment #1 by Reviewer #2).

- *The font size of the axis of Figure 2, 3 and 8 are too small. Please increase (to match the caption font size). Especially Figure 7 is really hard to read.*
  Figures have been modified.

[Figure]

**Figure 2**. Particle size distribution in winter (a) and summer (b) 2017, per each size bean mean circle, and median -, box bottom at 25%, box top at 75%, whisker bottom at 10%, and whisker top at 90%). Discontinuity at around 270 nm due to technical limitations of the UHSAS (handoff region between two internal gain stages).

[Figure]

**Figure 3.** Box and whisker plot of $N_{CCN,0.1\%}$ (a) and $N_{CCN,0.2\%}$ (b), mean $N_{CCN}$ blue circles, median -, box bottom at 25%, box top at 75%, whisker bottom at 10%, and whisker top at 90%, mean Ntot, black circles, and CCN potential activation fraction red circles.

[Figure]

**Figure 7.** Case study of December 2017 (leftmost), April 2017 (center), and September 2017 (rightmost) events. Submicron particle size distribution under baseline conditions (blue) and during the events (red), and $\kappa_{HTDMA}$ (open circles) during the events (a, f, k), Aitken, Accumulation, and Large Accumulation mode contributions to (b, g, l), non-refractory sulfate and organic aerosols (c, h, m), scatter plot of $N_{CCN}$ versus $N_{tot}$ during the event (red circle) and fitting lines for the events at SS 0.1% (red) and at SS 0.2% (dark red) (d, i, n), plot of potential activation ratio versus $N_{Ac} / N_{At}$, or the events at SS 0.1% (red) and at SS 0.2% (dark red) (e, j, o).

[Figure]

**Figure 8**. Mean percentage change in $N_{tot}$, $N_{At}$, $N_{Ac}$, $N_{CCN-0.1\%}$, and CCN potential activation fraction at SS 0.1% for each type of event (a); Aitken, Accumulation and Large Accumulation particle modes relative contribution to $N_{tot}$, for baseline and each type of event.

- *Figure 2: Add "technical limitations of the" before "UHSAS"*
  Added in the revised version of the manuscript.

    "**Figure 2**. Particle size distribution in winter (a) and summer (b) 2017, per each size bean mean circle, and median -, box bottom at 25%, box top at 75%, whisker bottom at 10%, and whisker top at 90%). Discontinuity at around 270 nm due to technical limitations of the UHSAS (handoff region between two internal gain stages)."

- *Figure 3: It is a bit difficult to see the activated fraction. Could you maybe change the color and connect the open circles with a line?*
  The figure has been modified.

[Figure]

**Figure 3.** Box and whisker plot of $N_{CCN,0.1\%}$ (a) and $N_{CCN,0.2\%}$ (b), mean $N_{CCN}$ blue circles, median -, box bottom at 25%, box top at 75%, whisker bottom at 10%, and whisker top at 90%, mean Ntot, black circles, and CCN potential activation fraction red circles.

- *Figure 4 to 6: It is really difficult to see any details of the CALIPSO trajectory (especially inFig 4). What do you actually like to show with these images? Maybe move them to the SI?*

CALIPSO trajectories have been included in the figures to show the passage of the satellite in respect to the ENA location. We thank Referee #2 for raising this criticism, and although we agree that showing CALIPSO trajectories is not strictly necessary, we feel it is a detailed that might be of interest to some readers and does not affect the clarity of the figures.

**Reference**

[revised manuscript text omitted]
_{4r}$ / $N_{4c}$) $N_{4r}$ / $N_{4c}$ change < 1% $N_{4r}$ contribution to $N_{tot}$ ~ 59% $N_{4r}$ contribution to $N_{tot}$ ~ 38% | Statistically significant shift in size ($N_{4r}$ / $N_{4c}$) $N_{4r}$ / $N_{4c}$ change > 200% $N_{4r}$ contribution to $N_{tot}$ ~ 42% $N_{4r}$ contribution to $N_{tot}$ ~ 56% | Statistically significant shift in size ($N_{4r}$ / $N_{4c}$) $N_{4r}$ / $N_{4c}$ change > 200% $N_{4r}$ contribution to $N_{tot}$ ~ 33% $N_{4r}$ contribution to $N_{tot}$ ~ 63% |
| Statistically non significant change in CCN potential Activation fraction $AF_{0.1\%}$ increase ~ 5%, $AF_{0.2\%}$ increase ~ 9% | Statistically significant change in CCN potential activation fraction $AF_{0.1\%}$ increase between 30% and 75% $AF_{0.2\%}$ increase between 35% and 100% | Statistically significant change in CCN potential activation fraction $AF_{0.1\%}$ and $AF_{0.2\%}$ increase > 75% |

Table 3. Summary of the characteristics of each type of multiday aerosol plume transport event. Underlined values indicate statistically significant Δ

| | **Dust and Marine mixture** | **Polluted continental and Marine mixture** | **Biomass Burning** |
|---|---|---|---|
| **Events Date (year 2017) and Origin** | • March 12 to 15 – Arctic/Canada
• November 26 to28 – North Africa
• December 07 to 10 – North Africa | • January 07 to 11 – North Europe
• April 20 to 22 – North Europe
• May 21 to 22 – North America
• October 11 to 13 – Hurricane | • August 26 to 29 – North America
• September 09 to 13 – North America |
| **Statistical analysis** | | | |
| $\Delta N_{tot}$ | > 110% | Between 95% and 110% | < 25% |
| $\Delta N_{At}/ N_{Ac}$ | < 1% | > 200% | > 200% |
| $\Delta AF_{0.1\%}$ | ~ 5% SS 0.1% | Between 30% and 75% | > 75% |
| $\Delta AF_{0.1\%}$ | ~ 7% SS 0.2% | Between 35% and 100% | > 75% |
| **Size mode fraction** | | | |
| $N_{At}$ contribution to $N_{tot}$ | ~ 59% | ~ 42% | ~ 33% |
| $N_{Ac}$ contribution to $N_{tot}$ | ~ 38% | ~ 56% | ~ 63% |

[Figure]

**Figure 1**. Box and whisker plot of monthly ubmicron aerosol number concentrations (box bottom at 25%, box top at 75%, whisker bottom at 10%, and whisker top at 90%). Mean (circles) and median (open circles) of total number concentration (black), number of Aitken (yellow), and  Accumulation (green) modes.

[Figure]

**Figure 2**. Particle size distribution in winter (a) and summer (b) 2017, per each size bean mean circle, and median -, box bottom at 25%, box top at 75%, whisker bottom at 10%, and whisker top at 90%). Discontinuity at around 270 nm due to technical limitations of the UHSAS (handoff region between two internal gain stages).

[Figure]

**Figure 3.** Box and whisker plot of $N_{CCN,0.1\%}$ (a) and $N_{CCN,0.2\%}$ (b), mean $N_{CCN}$ blue circles, median -, box bottom at 25%, box top at 75%, whisker bottom at 10%, and whisker top at 90%, mean $N_{tot}$, black circles, and CCN potential activation fraction red circles.

**Figure 4**. CALIPSO trajectories (blue) and Hysplit back trajectories (red) arriving at 50 m a.g.l. above the ENA site on December 07, 2017 (© Google Earth 2015).

[Figure]

**Figure 5.** CALIPSO trajectories (blue), and Hysplit back trajectories (red) arriving at 50 m a.g.l. above the ENA site on April 21, 2017 (© Google Earth 2015).

[Figure]

**Figure 6.** NASA Worldview VIIRS 375 Active fires between September 1 and 15, 2017 (red circles), and Hysplit back trajectories arriving at 50 m a.g.l. above the ENA site on September 10, 2017 (© Google Earth 2015).

[Figure]

[Figure]

**Figure 7.** Case study of December 2017 (leftmost), April 2017 (center), and September 2017 (rightmost) events. Submicron particle size distribution under baseline conditions (blue) and during the events (red), and $\kappa_{HTDMA}$ (open circles) during the events (a, , k), At, Ac, and LA mode contributions to (b, , l), non-refractory sulfate and organic aerosols (c, h, m), scatter plot of $N_{CCN}$ versus $N_{tot}$ during the event (red circle) and fitting lines for the events at SS 0.1% (red) and at SS 0.2% (dark red) (d, i, n), plot of potential activation ratio versus $N_{Ac}$ / $N_{At}$, or the events at SS 0.1% (red) and at SS 0.2% (dark red) (e, j, o).

[Figure]

**Figure 8**. Mean percentage change in $N_{tot}$, $N_{At}$, $N_{Ac}$, $N_{CCN-0.1\%}$, and CCN potential activation fraction at SS 0.1% for each type of event (a); Aitken, Accumulation and Large Accumulation particle modes relative contribution to $N_{tot}$, for baseline and each type of event.

---

## Author Response (AR2)

**Manuscript No.**: acp-2022-637

**Title**: Long-range transported continental aerosol in the Eastern North Atlantic: three multiday event regimes influence cloud condensation nuclei

**Responses to Anonymous Referee #2**

**1.[Referee #2]**: *Concerning the STP correction, I would like to mention that I was asking if the extensive parameters stated in the manuscript and presented in the figures (e.g. number concentrations) are given at standard temperature and pressure (usually 20 DegC and 1013 hPa) or at ambient conditions. The added sentence ("'Pressure for aerosol instruments is given at ambient conditions if not differently stated.") is not really addressing this.*

**[Resp.]:** To further clarify this point the sentence has been rewritten in the revised version of the manuscript and it reads as:

> **Page 4, Line 8**: "The pressure and temperature for the aerosol data is the same as reported in the ARM data archive which is given at ambient conditions to allow for direct comparision with the other data at ENA. If desired for future global comparion, data can be calculated at Standard Temperature and Pressure (STP) using the AOS meteorology data."

Reformating the data to Standard Temperature and Pressure (STP) would allow for comparison with other locations. However, the main focus of this manuscript is on ENA rather than global comparisons. We also did not report at STP here since different temperatures can be used for STP reporting. For example, the World Meteorological Organization and the Global Atmosphere Watch recommends reporting data at $0\ ^0C$ and 1013 hPa, rather than at $20\ ^0C$ as was mentioned here. For this reason, we thought this selection would be best made as is pertinent for the specific intercomparison to be made in future work.

**2.[Referee #2]**: *Concerning the calibration of HTDMA and CCNC: I think it is not sufficient to write "Typically, CCN counter and HTDMA instruments are calibrated once per year." and just reference instrumental hand-books. The authors should state the actual date/time of the calibrations, as well as check, and verify their data. For example, how did the calibration with ammonium sulfate or sodium chloride of HTDMA and CCNC look like? How was the sizing of the DMAs verified? This is an essential part of experimental work.*

**[Resp.]:** We appreciate Referee 2's concerns about instrument calibrations. We agree with Referee's 2 opinion that the calibration of the instruments is an essencial part of the experimental work. All data here is from the AOS at ENA, and all ARM AOS data is collected, ingested, and quality controlled by the U.S. DOE ARM User Facility, as indicated in Uin et al., 2019 and reported below for Referee 2's convenience:

- Uin, J., Aiken, A. C., Dubey, M. K., Kuang, C., Pekour, M., Salwen, C., Sedlacek, A. J., Senum, G., Smith, S., Wang, J., Watson, T. B., and Springston, S. R.: Atmospheric Radiation Measurement (ARM) Aerosol Observing Systems (AOS) for Surface-Based In Situ Atmospheric Aerosol and Trace Gas Measurements, J. Atmospheric Ocean. Technol., 36, 2429–2447, https://doi.org/10.1175/JTECH-D-19-0077.1, 2019

  > "Individual AOS instruments are operated under the direction of instrument mentors. Instrument mentors are experts of their respective instruments who provide the procurement specifications, perform or oversee calibrations and maintenance, work with the ARM Data Quality Office to automate (where possible) data quality assurance/quality control (QA/QC), develop documentation for and train site operators, and are the

ultimate authority for instrument data quality (Peppler et al. 2016). The AOS instrument mentors review deployment details and, if necessary, modify the written protocols for instrument operation.

Once a field site is set up, instrument mentors typically travel to the site for final integration, calibration and instrument verification. A one- to two-week period of data validation prior to the official start date allows time for data inspection and addressing unforeseen circumstances.

During the campaign, on-site operators fill in daily checklists with mentor-defined criteria for normal instrument response. Proper instrument operation is continually verified through data review by operators, mentors, the ARM Data Quality Office who have automated and manual review processes, or the principal investigator (PI) for the specific campaign who have nearly real-time access to processed data."

Peppler, RA, Kehoe, KE, Sonntag, KL, Bahrmann, CP, Richardson, SJ, Christensen, SW, McCord, RA, Doty, DJ, Wagener, Richard, Eagan, RC, Lijegren, JC, Orr, BW, Sisterson, DL, Halter, TD, Keck, NN, Long, CN, Macduff, MC, Mather, JH, Perez, RC, Voyles, JW, Ivey, MD, Moore, ST, Nitschke, DL, Perkins, BD, and Turner, DD. Quality Assurance of ARM Program Climate Research Facility Data. United States, doi:10.2172/948030, 2008

To address Referee #2's concerns, we added the following sentence and the reference to Peppler et al., 2008 and Uin et al. 2019 to the revised version of the manuscript.

> **Page 4, Line 11**: "All of the data is collected, ingested, quality controlled by the U.S. DOE ARM User Facility (Peppler et al., 2008, Uin et al., 2019)."

To further address instrument calibrations directly, calibration methodology is standardized for all ARM datasets and publicly reported in detail in the ARM instrument handbooks that can be found online. We cite the official ARM handbooks written by the individual instrument mentors for ARM in the manuscript in both Section 2 and Table 1. We also provide direct links to the documents in the reference section of the manuscript. For Referee 2's convenience, we report below the ARM handbook sections concerning CCN and HTMA calibration methodologies here.

- Uin, J.: Cloud Condensation Nuclei Counter (CCN) Instrument Handbook, 2016, https://www.arm.gov/publications/tech_reports/handbooks/ccn_handbook.pdf

> **6.5 Calibration Database**: During deployment the CCN is periodically calibrated by instrument mentors. CCN calibration involves generating and size-selecting ammonium sulfate particles and recording their total number concentration before and after activation in the CCN as a function of particle size. This is done for several humidifier column temperature gradients. Next, the 50% activation diameter (particle diameter where 50% of the generated ammonium sulfate particles are activated) is calculated for each humidifier column temperature gradient. As the activation characteristics of ammonium sulfate are known [2], supersaturation % can be calculated from the 50% activation diameter to establish/verify the relation between the temperature gradient of the humidifier column and the supersaturation %. Calibration coefficients are entered into a CCN configuration file and applied automatically by the instrument software. Coefficients are also recorded in the ARM Operation Status System (OSS, https://oss.arm.gov/oss.php) and archived by the instrument mentors."

> **12.0 Calibration**: The CCN is calibrated by the manufacturer before delivery to the user and during instrument maintenance at the manufacturer's facilities. The instrument mentors also typically perform calibration before and after each deployment at conditions (altitude) similar to the measurement site and during deployment if it has been more than a year from the last calibration and the deployment is not yet

ending. The calibration schedule is flexible because it depends on the availability of the calibration equipment (calibration scanning mobility particle sizer spectrometer [SMPS]). Calibration coefficients are entered into a CCN configuration file and applied automatically by the instrument software. Calibration results are also used by the mentors to assess the overall condition of the instrument, especially the humidifier column. Manufacturer's calibration includes:

- Supersaturation calibration with ammonium sulfate aerosol particles (see below).
- Size calibration of the OPC with National Institute of Standards and Technology (NIST)-traceable polystyrene latex (PSL) particles.
- Flow calibrations with a precision flow meter.

Mentor calibration of the CCN involves generating and size-selecting ammonium sulfate particles and recording their total number concentration before and after activation in the CCN as a function of particle size. This is done for several humidifier column temperature gradients. Next, the 50% activation diameter (particle diameter where 50% of the generated ammonium sulfate particles are activated) is calculated for each humidifier column temperature gradient. As the activation characteristics of ammonium sulfate are known [2], supersaturation % can be calculated from the 50% activation diameter to establish/verify."

[2] Rose, D, GP Frank, U Dusek, SS Gunthe, MO Andreae, and U Pöschl. 2007. "Calibration and measurement uncertainties of a continuous-flow cloud condensation nuclei counter (DMT-CCNC): CCN activation of ammonium sulfate and sodium chloride aerosol particles in theory and experiment." Atmospheric Chemistry and Physics Discussions 7(3): 8193−8260, http://doi.org/10.5194/acpd-7-8193-2007

- Uin, J.: 3002 Humidified Tandem Differential Mobility Analyzer Instrument Handbook, 2016, https://www.arm.gov/publications/tech_reports/handbooks/htdma_handbook.pdf

"**6.5 Calibration Data Base**: During deployment, the HT-DMA system is periodically (every six or twelve months) calibrated by the instrument mentors. This includes recording the growth factor of generated aerosol particles with a known chemical composition (ammonium nitrate or ammonium sulfate), as a function of RH. The growth factor value is compared to the Köhler model's theoretical values to ensure the HT-DMA is in a proper operational state. No correction factors are usually derived. Calibration data are collected and maintained by the instrument mentors."

"**12.0 Calibration**: The HT-DMA system is calibrated and validated by the manufacturer prior to delivery and during routine instrument maintenance tasks, performed at the manufacturing facilities. During deployment, subsets of the manufacturer's calibration are performed every six or twelve months by the instrument mentors. Manufacturer's calibration includes:

- Measuring the size distribution of a NIST traceable monodisperse calibration aerosol (PSL) to validate the proper operation of the DMAs.
- Measuring the growth factor of generated aerosol particles with a known chemical composition (ammonium sulfate or ammonium nitrate) as a function of RH, and comparing the resulting curve to a theoretical one from the Köhler model in order to validate the proper operation of the HT-DMA.
- Calibrating the high-voltage power supply, absolute pressure sensor, and laminar flow element against precision instruments and recording-derived correction factors in the software configuration files, which is used automatically.

Calibration and validation completed by the instrument mentors, similar to the manufacturer's calibration activities, includes:

- Recording the growth factor of generated aerosol particles with a known chemical composition (ammonium sulfate or ammonium nitrate), as a function of RH and comparing the resulting curve to a theoretical one from the Köhler model to validate the proper operation of the HT-DMA. If the recorded growth factor curve differs from the theoretical curve by more than the accuracy value specified by the manufacturer, it should be investigated on a case-by-case basis and corrected.
- Correction factors (if any) are derived according to the issue in question. Please consult the manufacturer's manual for more details."

Specifically, during the data period reported here, two calibrations (2016-09-29, and 2017-05-28) were performed by Janek Uin, a co-author on this manuscript and the instrument mentor for both instruments at ENA. He reported that there were no abnormal findings in the calibrations. We have added the following sentence to the text:

**Page 4, Line 13**: " "Calibrations were completed in accordance with the ARM instrument handbooks. No abnormalities were found during the periods that would affect the data reported here."

In summary, we have added clarifying sentences about how the data is handled in terms of data quality and instrument calibrations. We also report details in the references cited in the manuscript as shown above in case the reader would like further information and detail. Explicit description of instrument calibration methodologies and specific reports were not added to the manuscript since they are standardized by the ARM Facility as described and referenced above. Adding this information for all of the datasets reported from the AOS at ENA would add significant length to this manuscript, and we find this information to be beyond the scope of this manuscript since instrument calibrations are provided by the ARM Facility and references to those details can be found online.

References:

Peppler, RA, Kehoe, KE, Sonntag, KL, Bahrmann, CP, Richardson, SJ, Christensen, SW, McCord, RA, Doty, DJ, Wagener, Richard, Eagan, RC, Lijegren, JC, Orr, BW, Sisterson, DL, Halter, TD, Keck, NN, Long, CN, Macduff, MC, Mather, JH, Perez, RC, Voyles, JW, Ivey, MD, Moore, ST, Nitschke, DL, Perkins, BD, and Turner, DD. Quality Assurance of ARM Program Climate Research Facility Data. United States, doi:10.2172/948030, 2008

Uin, J., Aiken, A. C., Dubey, M. K., Kuang, C., Pekour, M., Salwen, C., Sedlacek, A. J., Senum, G., Smith, S., Wang, J., Watson, T. B., and Springston, S. R.: Atmospheric Radiation Measurement (ARM) Aerosol Observing Systems (AOS) for Surface-Based In Situ Atmospheric Aerosol and Trace Gas Measurements, J. Atmospheric Ocean. Technol., 36, 2429–2447, https://doi.org/10.1175/JTECH-D-19-0077.1, 2019

Uin, J.: 3002 Humidified Tandem Differential Mobility Analyzer Instrument Handbook, 2016, https://www.arm.gov/publications/tech_reports/handbooks/htdma_handbook.pdf

Uin, J.: Cloud Condensation Nuclei Counter (CCN) Instrument Handbook, 2016, https://www.arm.gov/publications/tech_reports/handbooks/ccn_handbook.pdf

WMO/GAW Aerosol Measurement Procedures, Guidelines and Recommendations, 2nd Edition. https://library.wmo.int/doc_num.php?explnum_id=3073, 2016.